

# Inferring subglacial topography using physics informed machine learning constrained by two conservation laws

Mansa Krishna[1], Gong Cheng[1], and Mathieu Morlighem[1]

[1]Department of Earth Sciences, Dartmouth College, Hanover, NH 03755, USA

**Correspondence:** Mansa Krishna (mansa.krishna.gr@dartmouth.edu)

**Abstract.** Subglacial topography beneath the Greenland Ice Sheet is a fundamental control on its dynamics and response to changes in the climate system. Yet, it remains challenging to measure directly, and existing representations of the subglacial topography rely on a limited number of observations. Although the use of mass conservation and the development of Bed-Machine Greenland substantially improved the representation of the bed topography, this approach is limited to fast-flowing sectors and is less effective in regions with complex, alpine topography. As an alternative to traditional numerical methods, recent work has explored using Physics Informed Neural Networks (PINNs), constrained by only one physical law, to solve forward and inverse problems in ice sheet modeling. Building on this work, we assess three PINN frameworks constrained by distinct conservation laws, showing that PINNs informed with a single conservation law are not sufficient for regions with sparse measurements and complex topographies. To that end, we introduce a novel approach that involves coupling *two* conservation laws within a PINN framework to infer the subglacial topography and test this approach for three regions with distinct environments in Greenland. This PINN is trained with both the conservation of mass and an approximation of the conservation of momentum (the Shelfy-Stream Approximation), which allows us to simultaneously infer the ice thickness and basal shear stress using observations of ice velocities, surface elevation, and the apparent mass balance in a mixed inversion problem. We compare the predicted ice thickness to ground-truth ice-penetrating radar measurements of ice thickness, showing that the PINN informed with two conservation laws is capable of inferring ice thickness in sparsely surveyed regions. Furthermore, comparisons of predicted bed topographies with BedMachine Greenland show that this approach is capable of discovering new bed features in slower-moving regions and in regions of complex topography, highlighting its potential for better constraining the bed topography of the Greenland Ice Sheet.

## 1 Introduction

Approximately 58 m and 7 m of sea level equivalent are sequestered in the Antarctic Ice Sheet and the Greenland Ice Sheet (GrIS), respectively (Morlighem et al., 2017, 2022, 2020; Morlighem, 2022), and they already contribute to global sea level rise at rates of 0.39 mm y$^{-1}$ and 0.74 mm y$^{-1}$, respectively (Otosaka et al., 2023). Despite an increase in the availability of remote sensing data and recent progress made in ice sheet numerical modeling (e.g., Nowicki et al., 2016; Goelzer et al., 2017, 2018; Nowicki et al., 2020; Seroussi et al., 2020, 2024), the uncertainty in the future mass balance of the ice sheets remains high (e.g., Aschwanden et al., 2021).



Among the sources of uncertainty, poorly constrained subglacial topography (or more simply, bed topography) remains one of the major factors affecting ice sheet model projections. While its impact at the ice sheet scale has not yet been rigorously quantified, Durand et al. (2011) showed that differences in the bed topography at resolutions greater than 2 km could lead to significant variations in ice sheet model behavior. More recently, Castleman et al. (2022) reported that the uncertainty in Thwaites Glacier's contribution to sea level due to the bed topography alone may be as large 21.9 cm over the next 200 years. The ice discharge at the ice sheet margin and stability of the grounding line and calving fronts are often preconditioned by the bed topography, with retrograde bed slopes having the potential to induce rapid retreat through the Marine Ice Sheet Instability (e.g., Schoof, 2012; Catania et al., 2018; Morlighem et al., 2019). In addition to controlling ice flow, the bed topography also influences the routing of subglacial water, which in turn affects basal sliding (e.g., Gagliardini et al., 2007; Schoof, 2010). Since the 1960s, ground-based and airborne ice-penetrating radars have been used to measure the ice thickness of the ice sheets, from which we can then obtain the bed elevation (e.g., Evans and Robin, 1966; Pritchard et al., 2025). Yet, these observations are limited in spatial coverage with gaps reaching tens of kilometers in some areas of the ice sheet, thus not meeting the ∼2 km spatial resolutions required by ice sheet models. Despite efforts to better constrain the bed topography through increased data acquisition (e.g., NASA's Operation IceBridge Paden et al., 2010, updated 2019), large uncertainties persist across many regions of the Antarctic and Greenland ice sheets.

Ice sheet models rely on indirect methods to resolve the bed topography between ice-penetrating radar-derived observations, which involve inferring the ice thickness (and subsequently the bed elevation) from observable ice sheet data. These data include surface elevation, ice velocities, surface and basal mass balances, and sparse ice thickness measurements from ice-penetrating radar. Traditional indirect methods involve using geo-statistical methods like kriging (e.g., Bamber et al., 2001, 2013) or numerically inverting for ice thickness using mass conservation (e.g., Morlighem et al., 2011). A combination of these traditional methods culminated in the development of BedMachine Greenland (Morlighem et al., 2017, 2022) and BedMachine Antarctica (Morlighem et al., 2020; Morlighem, 2022), which are currently widely used by the cryosphere community. Yet, the mass-conserving approach of BedMachine is limited to fast-flowing sectors and is often challenging to apply in regions with sparse measurements and complex topography, such as the mountainous areas of southeast Greenland.

Machine learning methods have risen as a promising new approach to improve upon existing representations of the bed and capture realistic bed topography features, with the ice thickness inversion framed as a "minimization problem" per the classification by Farinotti et al. (2017). However, standard machine learning methods may not necessarily account for the physical processes governing the dynamics of ice sheets and glaciers. Embedding prior knowledge, such as the fundamental conservation laws within a machine learning model architecture allows for the development of a robust framework (Raissi et al., 2019; Karniadakis et al., 2021). Such machine learning models, or neural networks, are called Physics-Informed Neural Networks (PINNs), which are designed to ensure that their predictions satisfy the governing physical principles (Raissi et al., 2019; Karniadakis et al., 2021). Within glaciology, PINNs were first used by Riel et al. (2021) to better understand the basal mechanics beneath glaciers and by Riel and Minchew (2023) to infer spatially-varying ice rheology. Since then, the literature on PINNs has grown (e.g., Bolibar et al., 2023; Iwasaki and Lai, 2023; Jouvet, 2023; Cheng et al., 2024; Wang et al., 2025).



In this paper, we use PINNs to infer the ice thickness, and subsequently the bed topography, in sectors of the GrIS where traditional inversion methods (such as the mass-conserving approach of BedMachine) are less effective. Following Cheng et al. (2024), we solve a two-dimensional, mixed inverse problem in which both the ice thickness and basal friction are inferred for three regions in Greenland. Our experiments explore different configurations of physical constraints within the PINN framework: mass conservation alone, momentum conservation alone, and a coupled system incorporating both conservation laws. The PINNs for each experiment are implemented using the open-source Python package, Physics-Informed Neural Networks for Ice and CLimatE (PINNICLE, Cheng et al., 2025b). Our objective is to determine the PINN framework and the physical constraints that are most effective for improving on existing representations of the bed topography in sectors with complex, alpine topography and slower-moving sectors further inland that do not have a high density of ice thickness observations.

## 2    Methods

The overall approach outlined in this section builds on the framework described in Cheng et al. (2024). We only provide a brief overview of the general PINN framework and our methods specific to this study, and refer the reader to Cheng et al. (2024, 2025b) for further details.

### 2.1    General Framework

We initialize and train the PINNs with the Python package, PINNICLE (Cheng et al., 2025b). The PINN inputs are the spatial coordinates and the outputs are the state variables in the partial differential equations (PDEs) used to constrain PINN predictions. Following the architecture described in Raissi et al. (2019), the PINN learns a relationship between the input and output variables from both the training data and from the PDEs with the help of a loss function, that describes the total error in PINN predictions, shown below:

$$\mathcal{L} = \mathcal{L}_{\text{data}} + \mathcal{L}_{\varphi}, \tag{1}$$

where $\mathcal{L}_{\text{data}}$ is the data loss and $\mathcal{L}_{\varphi}$ is the physical loss. Though the total loss can be defined in different ways (we refer the reader to Cheng et al. (2025b) for further details on loss function definitions), we use the mean squared error (MSE) to calculate the loss terms for this study. The data loss quantifies the misfit between PINN predictions and the training data. In particular, the PINN randomly selects a set of distinct spatial locations or "data points" within the region of interest and calculates the MSE between the PINN predictions and training data over these data points. The physical loss measures how well the PINN predictions satisfy the PDE residuals, thus serving as a soft constraint which enforces the physical laws. This is done by randomly selecting "collocation points" within the region of interest and calculating the MSE of the PDE residuals over these collocation points. During the training process, the PINN minimizes this loss function and updates its neural network coefficients accordingly, as illustrated in Fig. 1.





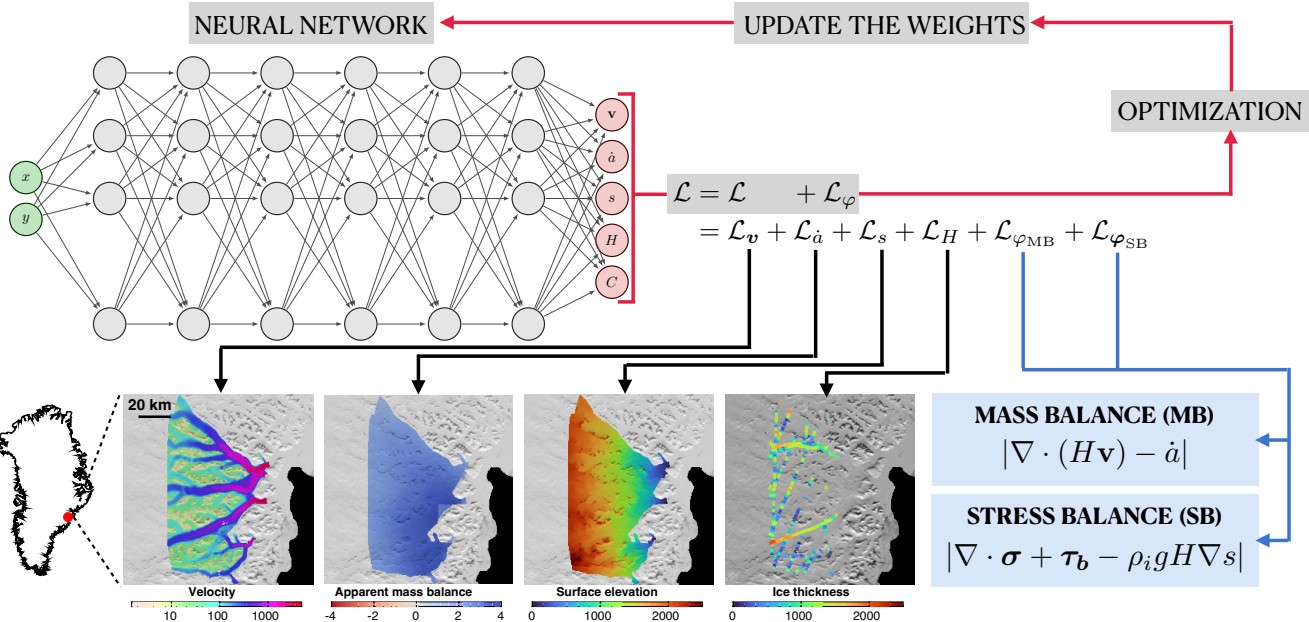

**Figure 1.** Illustration of the PINN framework, constrained by both mass conservation and momentum conservation, set up using the PINNI-CLE package. The inputs are the spatial coordinates and the outputs are the state variables of both PDEs. The loss function comprises of data loss terms (black arrows) and physical loss terms (blue arrows).

Our PINN framework (Fig. 1) uses 6 fully connected layers with 128 neurons each and a hyperbolic tangent activation function between layers. We use Adam optimization, a learning rate of $1.0 \times 10^{-4}$, and train the PINN for 700,000 iterations, after which the predictions do not improve significantly and training is considered sufficient.

## 2.2 Conservation Laws

Since we define our PINN output variables and select the training data according to the variables in the governing conservation laws, we first need to define the specific PDEs used for each of our experiments.

Let $\Omega \subset \mathbb{R}^2$ be the two-dimensional region of interest. The two-dimensional, depth-integrated conservation of mass for ice, considered an incompressible fluid, reads:

$$\frac{\partial H}{\partial t} + \nabla \cdot (H\boldsymbol{v}) = \dot{M}_s - \dot{M}_b, \tag{2}$$

such that for $\boldsymbol{x} = (x, y)^\mathsf{T} \in \Omega$, $H(\boldsymbol{x})$ is the ice thickness, $\boldsymbol{v}(\boldsymbol{x}) = (v_x, v_y)^\mathsf{T}$ is the depth-integrated ice velocity, $\dot{M}_b(\boldsymbol{x})$ is the basal melting rate, $\dot{M}_s(\boldsymbol{x})$ is the surface mass balance, and $\partial H(\boldsymbol{x})/\partial t$ is the ice thinning rate. Following Farinotti et al. (2009), we refer to the linear combination $\dot{a}(\boldsymbol{x}) = \dot{M}_s(\boldsymbol{x}) - \dot{M}_b(\boldsymbol{x}) - \partial H(\boldsymbol{x})/\partial t$ as the "apparent mass balance". Then, we denote the





mass balance residual, $\varphi_{\text{MB}}(\boldsymbol{x})$, as

$$\varphi_{\text{MB}} = \nabla \cdot (H\boldsymbol{v}) - \dot{a}. \tag{3}$$

For the conservation of momentum, we use the Shelfy-Stream/Shallow-Shelf Approximation (SSA, Morland, 1987; MacAyeal, 1989). Therefore, the stress balance residual, $\varphi_{\text{SB}}(\boldsymbol{x})$, is expressed as follows,

105 $$\varphi_{\text{SB}} = \nabla \cdot \boldsymbol{\sigma}_{\text{SSA}} + \boldsymbol{\tau_b} - \rho_i g H \nabla s, \tag{4}$$

where $\boldsymbol{\sigma}_{\text{SSA}}$ refers to the stress tensor, $\rho_i$ is the density of ice, $g$ is the acceleration due to gravity, and $s(\boldsymbol{x})$ is the surface elevation. $\boldsymbol{\tau_b}(\boldsymbol{x})$ refers to the basal shear stress calculated using Weertman's friction law (Weertman, 1957) shown below:

$$\boldsymbol{\tau_b} = -C^2 |\boldsymbol{v}|^{m-1} \boldsymbol{v} \tag{5}$$

where $C(\boldsymbol{x})$ represents the spatially-varying basal friction coefficient and $m = 1/3$. It should be noted that, in two-dimensions, 110 Eq. (4) consists of 2 equations, thus $\varphi_{\text{SB}}$ is a $(2 \times 1)$ vector.

### 2.3 Training Data

The training data include both direct measurements and reanalysis model outputs. More specifically, we provide the PINN with surface ice velocity, apparent mass balance, surface elevation, and sparse ice thickness measurements, all of which are variables in Eq. (3), (4), and shown in Fig. 2. Ice velocity data from NASA MEaSUREs products (Joughin et al., 2018) are 115 randomly selected for $N_{\boldsymbol{v}}$ distinct locations, $\{\boldsymbol{x}_j\}_{j=1}^{N_v} \in \Omega$, and denoted in tensor form, $\hat{\mathbf{v}}_{\text{data}} = (\hat{\boldsymbol{v}}_{\boldsymbol{x}\,\text{data}}, \hat{\boldsymbol{v}}_{\boldsymbol{y}\,\text{data}})^{\mathsf{T}}$. Apparent mass balance data from regional climate model RACMO 2.3 (Noël et al., 2016) combined with ICESat-2 derived ice thinning rates (Smith et al., 2020) are randomly selected for $N_{\dot{a}}$ distinct locations, $\{\boldsymbol{x}_j\}_{j=1}^{N_{\dot{a}}} \in \Omega$, and denoted in tensor form as $\hat{\dot{a}}_{\text{data}}$. Surface elevation data from the Greenland Ice Mapping Project (GIMP, Howat et al., 2014) are randomly selected for $N_s$ distinct locations, $\{\boldsymbol{x}_j\}_{j=1}^{N_s} \in \Omega$, and denoted in tensor form as $\hat{\boldsymbol{s}}_{\text{data}}$. Ice-penetrating radar measurements of ice thickness from CReSIS 120 radar depth sounder data (CReSIS, 2016) are randomly selected for $N_H$ distinct locations, $\{\boldsymbol{x_j}\}_{j=1}^{N_H} \in \Omega$, and denoted in tensor form as $\hat{\boldsymbol{H}}_{\text{data}}$. Unlike other training data, $\hat{\boldsymbol{H}}_{\text{data}}$ are only available along ice-penetrating radar flight tracks. Figure 2 depicts the aforementioned training data for a few regions in Greenland.

### 2.4 Regions of Interest

We focus on three different regions of Greenland in order to test the PINN frameworks in different environments as shown 125 in Fig. 2 and Fig. 3. We first focus on the region depicted in Fig. 2(a-d) in West Greenland, which includes Nunatakassaap Sermia and the three main branches of Upernavik Isstrøm, labelled in Fig. 3(a). For simplicity we will refer to this region as the "Upernavik" region. The bed topography in this region is well constrained as there are relatively dense ice thickness measurements from ice-penetrating radar, as shown in Fig. 2(d), making it appropriate for assessing the overall performance of the PINN. We then focus on a region in Southwest Greenland, where the ice velocities are slower, ice thickness measurements 130 are limited, and the bed topography is not well known. This "Narssap" region, shown in Fig. 2(e-h) includes Narssap Sermia,







**Figure 2.** Training data for three regions in Greenland. Each row corresponds to the training data for the (a-d) Upernavik, (e-h) Narssap, and (i-l) Deception regions. The first column on the left shows the location of these regions on the GrIS. The subsequent columns, from left to right, contain the following information: (a,e,i) magnitude of the ice velocity from NASA MEaSUREs, $|\hat{\mathbf{v}}_{\text{data}}|$ in $\log_{10}$ scale, (b,f,j) apparent mass balance derived using RACMO 2.3 and ICESat-2, $\hat{\dot{a}}_{\text{data}}$, (c,g,k) surface elevation from GIMP, $\hat{s}_{\text{data}}$, and (d,h,l) ice thickness $\hat{H}_{\text{data}}$, obtained along ice-penetrating radar flight tracks.





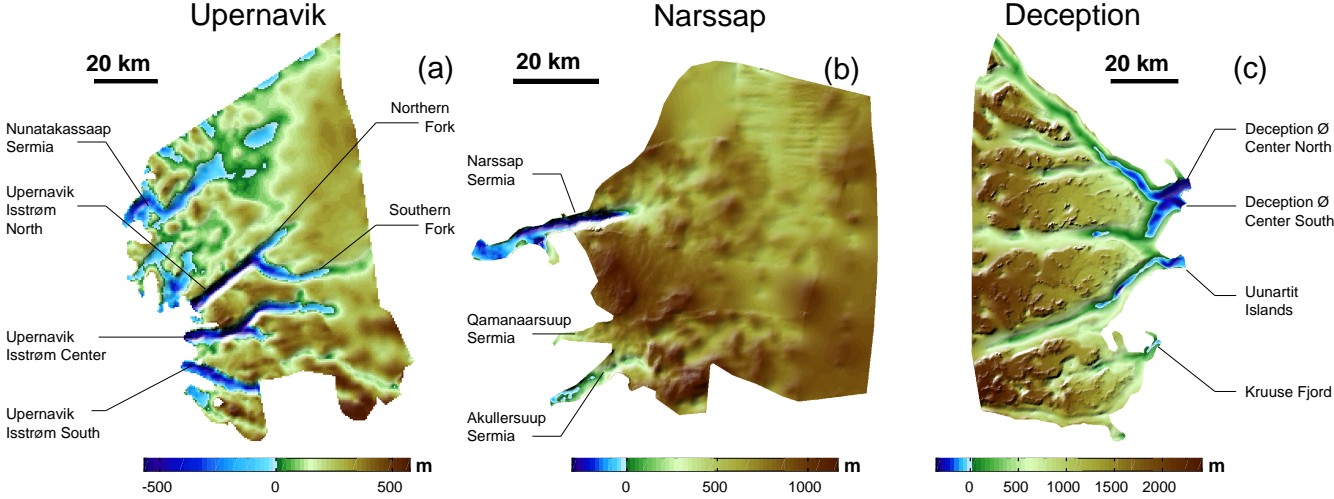

**Figure 3.** BedMachine Greenland bed topographies for (a) Upernavik, (b) Narssap, and (c) Deception regions, which we calculate using surface elevation from GIMP and BedMachine ice thickness. Each of the regions are annotated with the main outlet glaciers and fast-flowing ice streams.

Qamanaarsuup Sermia, and Akullersuup Semia, labelled in Fig. 3(b). Finally, we focus on the "Deception" region in Southeast Greenland, which has several fast-flowing, tributary ice streams, shown in Fig. 2(i-l). The main outlet glaciers are Unnamed Deception Ø Center North, Unnamed Deception Ø Center South, Unnamed Uunartit Islands, and Kruuse Fjord, which are labelled in Fig. 3(c). This region has sparse ice thickness measurements, a complex, alpine-like bed topography and slower ice velocities in between fast-flowing ice streams, making the mass-conserving approach of BedMachine difficult to implement (Morlighem et al., 2017).

## 2.5 Loss Function

Following the PINN architecture in Fig. 1, when a PINN constrained by two conservation laws is provided with input tensors $(\hat{\boldsymbol{x}}, \hat{\boldsymbol{y}})$, it will predict tensors of the state variables of both mass balance and stress balance, including $\hat{\mathbf{v}} = (\hat{\boldsymbol{v}}_{\boldsymbol{x}}, \hat{\boldsymbol{v}}_{\boldsymbol{y}})^{\mathsf{T}}$ (predicted ice velocity), $\hat{\boldsymbol{a}}$ (predicted apparent mass balance), $\hat{\boldsymbol{s}}$ (predicted surface elevation), $\hat{\boldsymbol{H}}$ (predicted ice thickness), and $\hat{\boldsymbol{C}}$ (predicted basal friction coefficient).

Let all $N_{(\cdot)}$ terms represent the number of data points associated with their corresponding subscripts (as defined in Section 2.2). We denote different $N_{(\cdot)}$ values for each variable because these data points are not necessarily in the same location as each other, which is standard practice when training PINNs (e.g., Cheng et al., 2024, 2025b). Then, supposing that $\|\hat{\boldsymbol{f}}\|^2 =$





$\Sigma_{j=1}^{N}(f_j)^2$ for any tensor $\hat{\boldsymbol{f}} = (f_1, \ldots, f_N)^{\mathsf{T}}$, we define the following data loss terms:

$$\mathcal{L}_{\boldsymbol{v}} = \frac{\gamma_{\boldsymbol{v}}}{N_{\boldsymbol{v}}} \|\hat{\boldsymbol{v}}_{\boldsymbol{x}\,\mathrm{data}} - \hat{\boldsymbol{v}}_{\boldsymbol{x}}\|^2 + \frac{\gamma_{\ln \boldsymbol{v}}}{N_{\boldsymbol{v}}} \left\| \ln \frac{|\hat{\boldsymbol{v}}_{\boldsymbol{x}\,\mathrm{data}}| + \varepsilon}{|\hat{\boldsymbol{v}}_{\boldsymbol{x}}| + \varepsilon} \right\|^2 + \frac{\gamma_{\boldsymbol{v}}}{N_{\boldsymbol{v}}} \|\hat{\boldsymbol{v}}_{\boldsymbol{y}\,\mathrm{data}} - \hat{\boldsymbol{v}}_{\boldsymbol{y}}\|^2 + \frac{\gamma_{\ln \boldsymbol{v}}}{N_{\boldsymbol{v}}} \left\| \ln \frac{|\hat{\boldsymbol{v}}_{\boldsymbol{y}\,\mathrm{data}}| + \varepsilon}{|\hat{\boldsymbol{v}}_{\boldsymbol{y}}| + \varepsilon} \right\|^2, \tag{6}$$

$$\mathcal{L}_{\dot{a}} = \frac{\gamma_{\dot{a}}}{N_{\dot{a}}} \|\hat{\dot{\boldsymbol{a}}}_{\mathrm{data}} - \hat{\dot{\boldsymbol{a}}}\|^2, \tag{7}$$

$$\mathcal{L}_s = \frac{\gamma_s}{N_s} \|\hat{\boldsymbol{s}}_{\mathrm{data}} - \hat{\boldsymbol{s}}\|^2, \tag{8}$$

$$\mathcal{L}_H = \frac{\gamma_H}{N_H} \|\hat{\boldsymbol{H}}_{\mathrm{data}} - \hat{\boldsymbol{H}}\|^2, \tag{9}$$

where the data loss terms $\mathcal{L}_{(\cdot)}$ correspond to the MSE with respect to their subscripts (as defined in Section 2.2). The velocity data loss term $\mathcal{L}_{\boldsymbol{v}}$ shown in Eq. (6) includes two additional logarithmic terms that allow the PINN to capture slower ice velocities. These logarithmic terms include a small number $\varepsilon \sim 10^{-16}$ to prevent from taking the natural log of zero or diving by zero. Given the framework depicted in Fig. 1, we note that the PINN is required to predict the basal friction coefficient in order to compute the stress balance residual, however we do not include a basal friction data loss term in the overall loss

function since we do not have direct observations of the same.

For the physical loss terms, we randomly select $N_{\varphi}$ collocation points $\{\boldsymbol{x}_j\}_{j=1}^{N_{\varphi}} \in \Omega$ to compute mass balance and stress balance shown in Eq. (3) and (4) respectively. In particular, we denote tensors of the mass balance and stress balance residuals at these collocation points as $\hat{\boldsymbol{\varphi}}_{\mathrm{MB}}$ and $\hat{\boldsymbol{\varphi}}_{\mathrm{SB}} = (\hat{\boldsymbol{\varphi}}_{\mathrm{SB}_x}, \hat{\boldsymbol{\varphi}}_{\mathrm{SB}_y})^{\mathsf{T}}$ respectively. Then, we define the following physical loss terms:

$$\mathcal{L}_{\boldsymbol{\varphi}_{\mathrm{MB}}} = \frac{\gamma_{\mathrm{MB}}}{N_{\varphi}} \|\hat{\boldsymbol{\varphi}}_{\mathrm{MB}}\|^2, \tag{10}$$

$$\mathcal{L}_{\boldsymbol{\varphi}_{\mathrm{SB}}} = \frac{\gamma_{\mathrm{SB}}}{N_{\varphi}} \|\hat{\boldsymbol{\varphi}}_{\mathrm{SB}_x}\|^2 + \frac{\gamma_{\mathrm{SB}}}{N_{\varphi}} \|\hat{\boldsymbol{\varphi}}_{\mathrm{SB}_y}\|^2. \tag{11}$$

where the $\mathcal{L}_{\boldsymbol{\varphi}_{(\cdot)}}$ terms correspond to the MSE of the PDE residuals with respect to their subscripts: MB (Mass Balance) and SB (Stress Balance).

| Variable(s) | Typical Value | Weighting Term | Weight Value | Number of Points |
|---|---|---|---|---|
| $\|\boldsymbol{v}\|$ | $10^4 \, \mathrm{m\,y^{-1}}$ | $\gamma_{\boldsymbol{v}}$ | $10^{-8} \times 31536000^2 \, \mathrm{m^{-2}\,s^2}$ | $N_{\boldsymbol{v}} = 4000$ |
| $\ln\|\boldsymbol{v}\|$ | $10^1 \, \mathrm{m\,y^{-1}}$ (in $\log_e$ scale) | $\gamma_{\ln \boldsymbol{v}}$ | $10^{-5} \, \mathrm{m^{-2}\,s^2}$ (in $\log_e$ scale) | $N_{\boldsymbol{v}} = 4000$ |
| $\dot{a}$ | $1 \, \mathrm{m\,y^{-1}}$ | $\gamma_{\dot{a}}$ | $31536000^2 \, \mathrm{m^{-2}\,s^2}$ | $N_{\dot{a}} = 4000$ |
| $s, H$ | $10^3 \, \mathrm{m}$ | $\gamma_s, \gamma_H$ | $10^{-6} \, \mathrm{m^{-2}}$ | $N_s = N_H = 4000$ |
| $\rho_i g H \nabla s$ | $10^5 \, \mathrm{Pa}$ | $\gamma_{\mathrm{SB}}$ | $10^{-12} \, \mathrm{Pa^{-2}}$ | $N_{\varphi} = 9000$ |
| $\nabla \cdot (H\boldsymbol{v})$ | $1 \, \mathrm{m\,y^{-1}}$ | $\gamma_{\mathrm{MB}}$ | $10^{12} \, \mathrm{m^{-2}\,s^2}$ | $N_{\varphi} = 9000$ |

**Table 1.** Typical values of variables, selected weighting terms for the loss function, and number of data points and collocation points chosen for all three regions

All the $\gamma_{(\cdot)}$ coefficients represent carefully selected scaling terms, or loss function weights, following a similar strategy to Cheng et al. (2024). All $\gamma_{(\cdot)}$ weights ensure that each of the loss terms contribute similarly to the overall loss within the SI unit



system. Both the data loss and physical loss terms are scaled to roughly the same order of magnitude using the typical values
of variables in ice sheet modeling applications, as shown in Table 1. We note that there is some flexibility in the choice of
these weighting terms as the PINN output variables will have different values for different regions of the GrIS. Indeed, the $\gamma_{(\cdot)}$
coefficients presented in this paper differ slightly than those provided by Cheng et al. (2024, 2025b). Table 1 provides further
details on the typical values of these variables, the corresponding weights, and the number of data points and collocation points
chosen for training.

## 2.6  Experiments

For each of the three experiments, described below, we train the same PINN multiple times to obtain a *median* prediction for
each of the PINN output variables. This approach is necessary because the mixed inverse problem is ill-posed and the PINN
training process is sensitive to randomness. More specifically, the initial state of the PINN (i.e., the initial neural network
weights), the locations of collocation points, and the selection of data points are all determined randomly, implying that the
final PINN predictions for each region may vary. As such, we train five PINNs for each experiment with different random seeds
and retrieve the *median* prediction over a given region of interest.

### 2.6.1  Mass Balance

The goal of this experiment is to assess how well PINNs perform in regions where the mass-conserving approach of BedMa-
chine is challenging to implement. Hence, we train PINNs constrained with only the mass balance residual shown in Eq. (3)
for Deception, which is a complex region with sparse measurements and slow-moving ice in between fast-flowing ice streams.
Henceforth, we will refer to this PINN as "PINN (MB)" (or PINN constrained with Mass Balance only).

The architecture of the PINN (MB) differs slightly from the architecture illustrated in Fig. 1. When provided with input
tensors $(\hat{\boldsymbol{x}}, \hat{\boldsymbol{y}})$, PINN (MB) will predict output tensors of the state variables of the conservation of mass. Therefore, we construct
the following loss function, denoted by $\mathcal{L}_{\mathrm{MB}}$, to inform predictions:

$$\mathcal{L}_{\mathrm{MB}} = \mathcal{L}_{\boldsymbol{v}} + \mathcal{L}_{\dot{a}} + \mathcal{L}_H + \mathcal{L}_{\varphi_{\mathrm{MB}}} \tag{12}$$

where all the loss terms $\mathcal{L}_{(\cdot)}$ are defined in Eq. (6), (7), (9), and (10). We retain all $N_{(\cdot)}$ and $\gamma_{(\cdot)}$ values from Table 1 for
PINN (MB). We do not explicitly include boundary conditions in the loss function for this experiment. Instead, the boundary
conditions are implicitly defined as PINN (MB) is exposed to ice thickness data along ice-penetrating radar flight tracks within
the Deception region, as shown in Fig. 2(l).

### 2.6.2  Stress Balance

Recent work by Cheng et al. (2024) demonstrated the application of PINNs constrained with only the conservation of mo-
mentum to the fast-flowing region of Helheim Glacier. However, it remains unclear whether this PINN framework would be as
effective as BedMachine in regions with more complex topography, sparse measurements, and slower ice velocities. To explore



this further, we implement a PINN constrained with only the stress balance residual shown in Eq. (4) for the Deception region. We refer to this PINN as "PINN (SB)" (or PINN constrained with Stress Balance only).

Similar to PINN (MB), the architecture of PINN (SB) also differs slightly from the architecture illustrated in Fig. 1. When provided with input tensors $(\hat{\boldsymbol{x}}, \hat{\boldsymbol{y}})$, PINN (SB) will predict output tensors of the state variables of the conservation of momentum. Therefore, we construct the following loss function, denoted by $\mathcal{L}_{\mathrm{SB}}$, to inform predictions:

$$\mathcal{L}_{\mathrm{SB}} = \mathcal{L}_{\boldsymbol{v}} + \mathcal{L}_s + \mathcal{L}_H + \mathcal{L}_{\varphi_{\mathrm{SB}}} \tag{13}$$

where all the loss terms $\mathcal{L}_{(\cdot)}$ are defined in Eq. (6), (8), (9), and (11). We retain all $N_{(\cdot)}$ and $\gamma_{(\cdot)}$ values from Table 1 for PINN (SB). Cheng et al. (2024) included additional loss terms describing boundary conditions on the calving front of Helheim Glacier, however since we are only solving the inverse problem (and not the forward problem), we do not include any loss terms for calving. Cheng et al. (2024) also included Dirichlet boundary conditions for the remaining boundaries, however for

this paper, the boundary conditions are implicitly defined. More specifically, PINN (SB) is exposed to the ice velocity data along the boundaries of the region of interest, thereby satisfying the stress balance boundary conditions.

### 2.6.3   Coupling Mass Balance and Stress Balance

The results of BedMachine Greenland and Cheng et al. (2024) suggest that that informing PINNs with mass conservation or momentum conservation alone may not be sufficient to retrieve accurate predictions of ice thickness in slower-moving regions

with sparse ice thickness measurements and complex topographies. This PINN framework, depicted in Fig. 1, allows us to easily and flexibly introduce multiple PDE constraints within our loss function. Hence, we implement a PINN constrained with both mass balance, Eq. (3), and stress balance, Eq. (4), for the regions of interest, described above. These PDEs are coupled within the PINN framework as predicted ice thickness values that satisfy the mass balance residual and directly substituted into the stress balance residuals, and vice versa. Henceforth, we refer to the PINN in this experiment as "PINN (MB+SB)" (or

PINN constrained with both Mass Balance and Stress Balance).

As shown in Fig. 1, when this PINN (MB+SB) is provided with input tensors $(\hat{\boldsymbol{x}}, \hat{\boldsymbol{y}})$, it will predict output tensors of the state variables of both the conservation of mass and the conservation of momentum. Therefore, we construct the following loss function, denoted by $\mathcal{L}$, to inform predictions:

$$\mathcal{L} = \mathcal{L}_{\boldsymbol{v}} + \mathcal{L}_{\dot{a}} + \mathcal{L}_s + \mathcal{L}_H + \mathcal{L}_{\varphi_{\mathrm{MB}}} + \mathcal{L}_{\varphi_{\mathrm{SB}}} \tag{14}$$

where the loss terms $\mathcal{L}_{(\cdot)}$ are defined in Eq. (6), (7), (8), (9), (10), and (11). We retain all $N_{(\cdot)}$ and $\gamma_{(\cdot)}$ values from Table 1 for this PINN (MB+SB). As described in the previous experiments, we do not explicitly include loss terms for the boundary conditions in this experiment, rather these are defined implicitly.

### 3   Results

Depending on the configuration of PDE constraints, each PINN took approximately 3-17 hours to train using the NVIDIA

A100 PCIE 40GB GPU (Texas Advanced Computing Center). We train five PINNs for each experiment with different random





seeds to deal with the random nature of the PINN training process. Unless otherwise stated, all reported results refer to the *median* prediction of five PINNs.

## 3.1 Effect of Different Physical Constraints

To assess the effect of different physical constraints, we present PINN (MB), PINN (SB), and PINN (MB+SB) predictions of

the bed topography for Deception. For each experiment, we use predictions of the ice thickness, $\hat{H}$, and the surface elevation from GIMP, $\hat{s}_{\mathrm{data}}$, to calculate the predicted bed topography, $\hat{b} = \hat{s}_{\mathrm{data}} - \hat{H}$, as depicted in Fig. 4(a-c). The differences between predicted bed topographies and BedMachine Greenland are depicted in Fig. 4(d-f), with the areas in which BedMachine applies a mass-conserving approach outlined in black; henceforth we will refer to these outlined areas as $\Omega_{\mathrm{mc}}$.

| Model | PINNs vs. $b_{\mathrm{data}}$ (m) | | |
|---|---|---|---|
| | $\Omega$ | $\Omega_{\mathrm{mc}}$ | $\Omega \setminus \Omega_{\mathrm{mc}}$ |
| BedMachine | 324 | 76.2 | 392 |
| PINN (MB) | 123 | 98.5 | 133 |
| PINN (SB) | 97.5 | 71.3 | 108 |
| PINN (MB+SB) | 137 | 105 | 150 |

**Table 2.** Bed topography RMSEs for Deception, for different physical constraints.

To assess the accuracy of the predicted bed topographies, we compare them to 'ground-truth data' obtained along ice-

penetrating radar flight tracks. We compute these reference bed elevations by taking the difference between observed surface elevation and ice thickness, $\hat{b}_{\mathrm{data}} = \hat{s}_{\mathrm{data}} - \hat{H}_{\mathrm{data}}$. The root mean squared errors (RMSEs) between the inferred and reference bed topographies are compiled into Table 2.

The BedMachine and PINN (MB) topographies are quite different as shown in Fig. 4(d), especially outside of $\Omega_{\mathrm{mc}}$ where differences exceed 800 m in some areas. PINN (MB) predictions align more closely with the ground-truth bed topography, as

indicated by the relatively low RMSE of 123 m compared to the BedMachine RMSE of 324 m (see Table 2). However, we also notice that PINN (MB) predicts bed features that are highly correlated to the ice-penetrating radar flight tracks, shown in Fig. 2(l). Within $\Omega_{\mathrm{mc}}$, the average difference between BedMachine and PINN (MB) is significantly reduced, with differences of approximately 300 m. Yet, PINN (MB) tends to predict thinner ice than ice-penetrating radar measurements suggest. As a result, PINN (MB) does not capture the depth of the troughs beneath Deception Ø Center North, Deception Ø Center South

and Uunartit Islands that are better resolved by BedMachine. Indeed, the PINN (MB) RMSE within $\Omega_{\mathrm{mc}}$ of 98.5 m is higher than the BedMachine RMSE of 76.2 m (see Table 2).

For PINN (SB), we notice that the predicted bed topography for Deception is relatively similar to BedMachine outside $\Omega_{\mathrm{mc}}$, as shown by the grey areas on the difference map in Fig. 4(e). More quantitatively, the PINN (SB) RMSE of 108 m outside $\Omega_{\mathrm{mc}}$ is significantly lower than the BedMachine RMSE of 392 m outside $\Omega_{\mathrm{mc}}$ (see Table 2). We observe that PINN (SB)

predicts isolated crater-like depressions along ice-penetrating radar flight tracks, shown in Fig. 4(b). Within $\Omega_{\mathrm{mc}}$, the difference



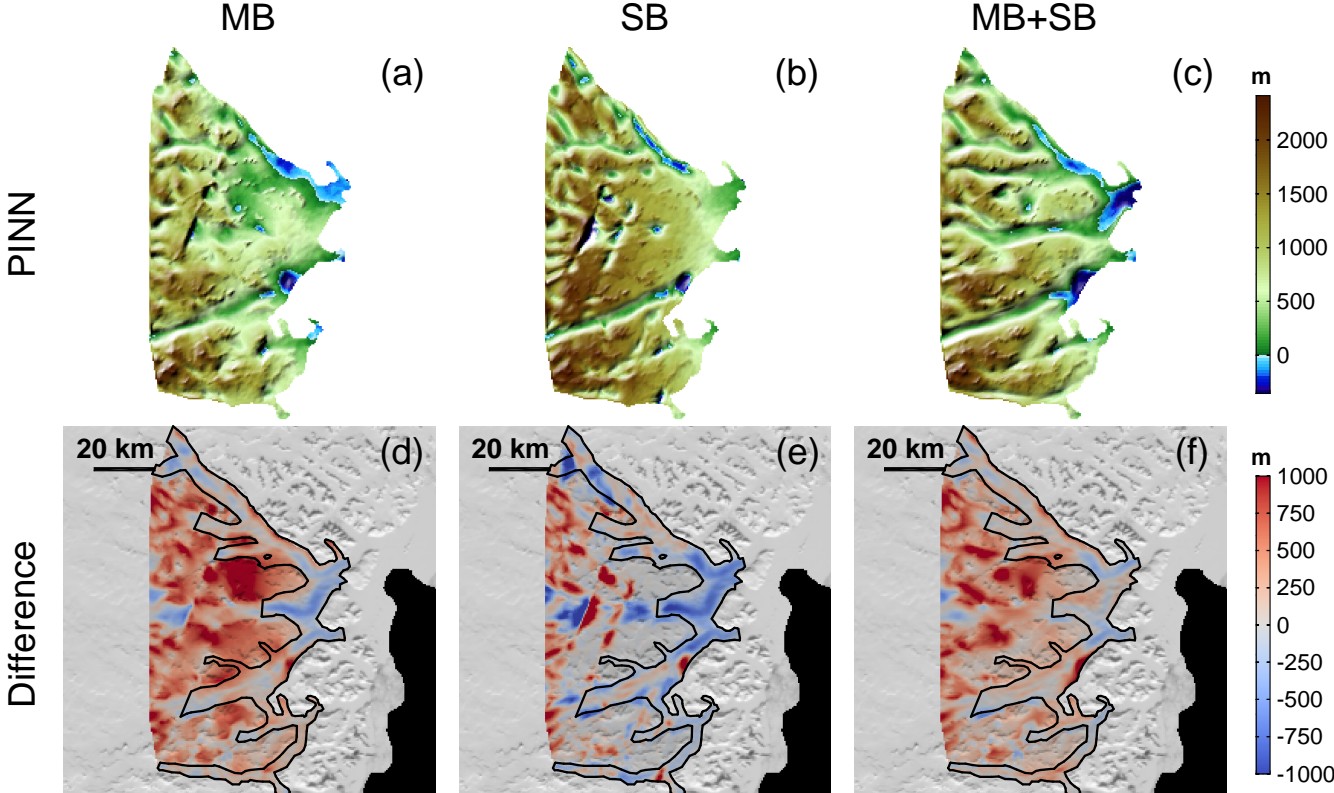

**Figure 4.** Inferred bed topography maps for Deception to demonstrate the effect of different physical constraints. The depicted maps are for (a,d) PINN (MB), (b,e) PINN (SB), and (c,f) PINN (MB+SB). From top to bottom, (a-c) show the PINN inferred bed topography maps, and (d-f) show the difference between BedMachine and PINN bed topography maps. On these difference maps, we outline the areas in which BedMachine uses a mass-conserving approach, $\Omega_{mc}$ (i.e., the black outline).

between BedMachine and PINN (SB) is more pronounced, with differences of approximately 600 m. Interestingly, though the PINN (SB) RMSE of 71.3 m is similar to the BedMachine RMSE of 76.2 m within $\Omega_{mc}$, PINN (SB) predicts thinner ice than ice-penetrating radar measurements suggest and fails to capture the troughs beneath Deception Ø Center North, Deception Ø Center South and Uunartit Islands that are well captured in BedMachine.

Lastly, we observe that the Bedmachine and PINN (MB+SB) bed topographies are significantly different. However, unlike PINN (MB) and PINN (SB), which predict isolated bed features that coincide with radar flight tracks, the PINN (MB+SB) bed topography for Deception consists of an intricate network of troughs that are not well captured in BedMachine. PINN (MB+SB) appears to connect and interpolate bed features between radar flight tracks, where we have direct observations. Though we observe differences of over 600 m outside $\Omega_{mc}$, as shown in Fig. 4(f), the PINN (MB+SB) RMSE of 137 m is

lower than that of BedMachine (see Table 2). Within $\Omega_{mc}$, PINN (MB+SB) does capture the troughs beneath Deception Ø Center North, Deception Ø Center South and Uunartit Islands. However, the predicted ice thickness is slightly thinner than





suggested by radar-derived measurements and BedMachine. As a final note, we determine that PINN (MB+SB) produces accurate predictions through a separate validation study and refer the reader to Appendix C.

## 3.2 Predictions for Regions of Interest

Moving forward, we present the applications of PINN (MB+SB) to the three regions of interest: Upernavik, Narssap, and Deception. Further detail on our results, including more detail on predicted basal friction coefficients, can be found in Appendix A.

### 3.2.1 Predictions of the Bed Topography

We calculate the predicted bed topography, $\hat{b}$, for each region as shown in Fig. 5(a-c) and compare these bed topographies to the 270 BedMachine bed topographies shown in Fig. 3. We show the difference between the BedMachine and PINN bed topographies in Fig. 5(d-f), annotated with $\Omega_{\text{mc}}$. To assess the PINN results, we compare these to ground-truth bed topography data, $\hat{b}_{\text{data}}$, and compile the RMSEs into in Table 3.

| Region | PINN vs. $b_{\text{data}}$ (m) | | | BedMachine vs. $b_{\text{data}}$ (m) | | |
|---|---|---|---|---|---|---|
| | $\Omega$ | $\Omega_{\text{mc}}$ | $\Omega \setminus \Omega_{\text{mc}}$ | $\Omega$ | $\Omega_{\text{mc}}$ | $\Omega \setminus \Omega_{\text{mc}}$ |
| Upernavik | 70.6 | 70.6 | - | 93.1 | 93.1 | - |
| Narssap | 52.7 | 48.2 | 63.5 | 79.2 | 74.9 | 89.9 |
| Deception | 137 | 105 | 150 | 324 | 76.2 | 392 |

**Table 3.** Bed topography RMSEs for Upernavik, Narssap, and Deception along ice-penetrating radar tracks. The table contains RMSEs between PINN-predicted bed topographies and ground-truth ice-penetrating radar observations, $\hat{b}_{\text{data}}$ (m), both within and outside $\Omega_{\text{mc}}$. For comparison, we also include RMSEs between BedMachine bed topographies and ground-truth ice-penetrating radar observations within and outside $\Omega_{\text{mc}}$.

For Upernavik, the BedMachine and PINN bed topographies, shown in Fig. 3(a) and Fig. 5(a) respectively, are broadly similar with an average difference of less than 100 m for the whole region. The PINN RMSE of 70.6 m (compared to ground-275 truth bed topography data) is lower than the BedMachine RMSE of 93.1 m (see Table 3), and indeed, we observe a few noticeable differences between these bed topographies. Going from north to south, the PINN predicts a continuous trough beneath Nunatakassaap Sermia, while BedMachine does not. Second, the PINN indicates that the trough beneath the northern fork of Upernavik Isstrøm North extends further inland than BedMachine suggests. Lastly, the PINN predicts a 'disconnected' trough beneath the southern fork of Upernavik Isstrøm North, while BedMachine suggests that this trough is continuous.

Comparing the PINN bed topographies for Narssap in Fig. 5(b) and BedMachine in Fig. 3(b), we observe that the PINN infers glacial valleys in the bed, especially outside $\Omega_{\text{mc}}$. These features satisfy ground-truth bed topography data as indicated by the relatively low RMSE of 52.7 m compared to the BedMachine RMSE of 79.2 m (see Table 3). Within $\Omega_{\text{mc}}$, the PINN predicts





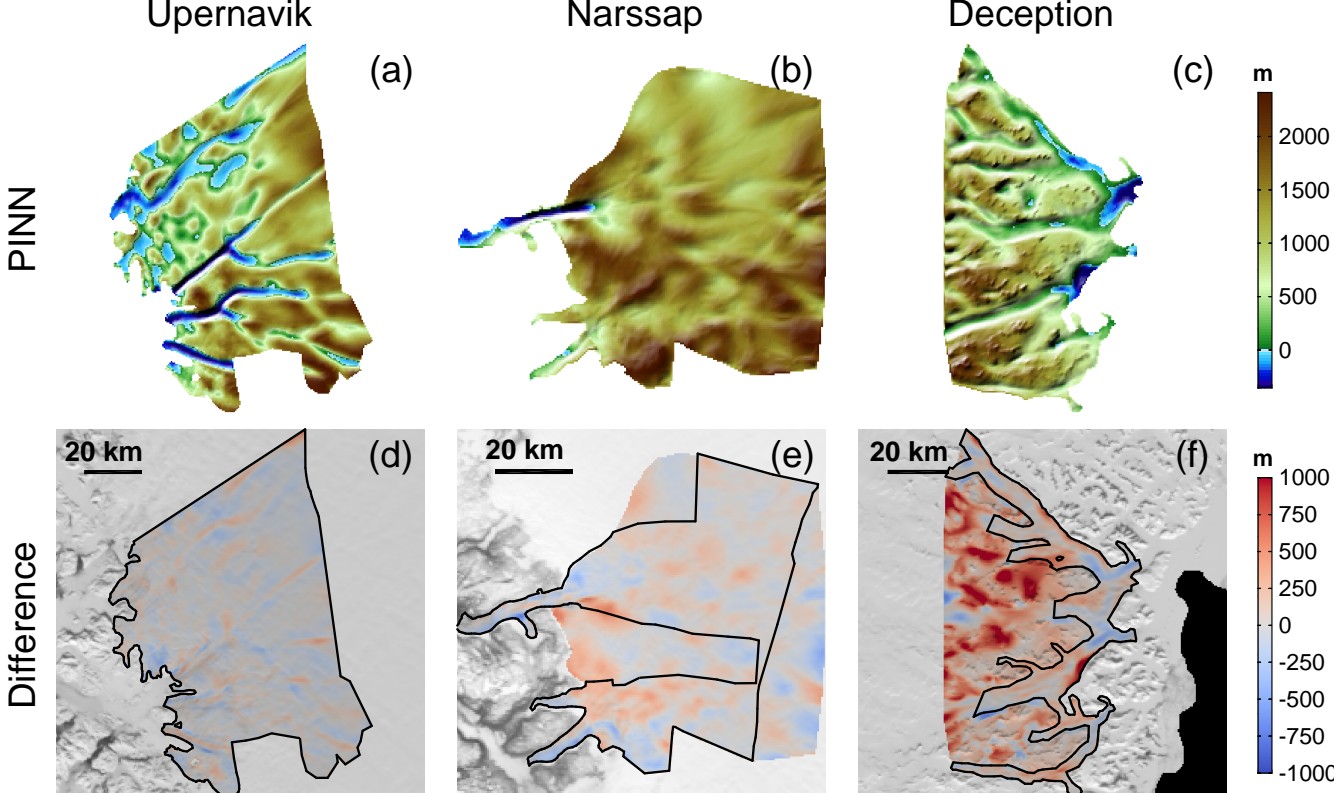

**Figure 5.** PINN (MB+SB) bed topography and difference maps for (a,d) Upernavik, (b,e) Narssap, and (c,f) Deception. (a-c) depict the PINN (MB+SB) predicted bed topography maps, and (d-f) depict the difference between BedMachine and the PINN bed topography maps. These difference maps are also annotated with an outline of $\Omega_{mc}$ for each region (in black).

troughs beneath Narssap Sermia and Akullersuup Sermia that are shallower than those in both BedMachine and ground-truth data by approximately 200 m.

The BedMachine and PINN bed topographies for Deception, shown in Fig. 3(c) and Fig. 5(c) are significantly different. The PINN predicts an intricate network of troughs that coincide with the fast-flowing ice streams and are not well captured in BedMachine. These differences are especially evident outside $\Omega_{mc}$ in Fig. 5(f), with the PINN showing that the bed topography is deeper than that of BedMachine by over 600 m. Despite this rather large difference with respect to BedMachine, these PINN inferred bed features satisfy ground-truth bed topography data, as indicated by the relatively low RMSE of 150 m compared
to the BedMachine RMSE of 392 m outside $\Omega_{mc}$ (see Table 3). Within $\Omega_{mc}$, the PINN predicts troughs beneath Deception Ø Center South and Uunartit Islands that are shallower than that of both BedMachine and ground-truth data by approximately 250 m.





### 3.2.2 Predictions of Observable Ice Sheet Features

We present PINN predictions of the observable ice sheet features (including the ice velocity, apparent mass balance, and surface
elevation) for all three regions of interest. It is important that the PINN accurately predicts these observable features as these
predictions are used to compute the PDE residuals in the loss function, and therefore the bed prediction. Table 4 describes the
RMSEs between PINN predictions and the training data for these observable features.

| Glacier | $|\boldsymbol{v}|$ (m y$^{-1}$) | $\dot{a}$ (m y$^{-1}$) | $s$ (m) |
|---|---|---|---|
| Upernavik | 40.7 | 0.112 | 20.9 |
| Narssap | 24.8 | 0.0209 | 12.3 |
| Deception | 62.8 | 0.0321 | 56.2 |

**Table 4.** RMSEs between PINN predictions and the training data for each region

The PINNs are generally able to capture the surface ice velocities, though predictions have larger differences of over 100 m
y$^{-1}$ along fast-flowing ice streams, as shown in Fig. A1. Upernavik and Narssap velocity predictions have lower RMSEs of
40.7 m y$^{-1}$ and 24.8 m y$^{-1}$ respectively, while the Deception velocity prediction has a higher RMSE of 62.8 m y$^{-1}$, which
is to be expected since Deception has faster-flowing glaciers. Among all the observable features, the PINNs best capture the
apparent mass balance within $\pm 0.5$ m y$^{-1}$ for each region (shown in Table 4), though we notice minor differences along the
margins, as shown in Fig. A2. The Narssap and Deception apparent mass balance predictions have lower RMSEs of 0.0209 m
y$^{-1}$ and 0.0321 m y$^{-1}$ respectively, while the RMSE of the apparent mass balance in Upernavik is an order of magnitude higher
at 0.112 m y$^{-1}$. The PINNs also generally capture the surface elevation for each glacier, though these predictions are smoother,
implying that they are lower frequency approximations of the observed surface elevation from GIMP, as shown in Fig. A3. In
other words, the PINNs do not resolve fine-scale ($\lesssim 500$ m) details in the surface elevation. The surface elevation predictions
for Upernavik and Narssap have lower RMSEs of 20.9 m and 12.3 m respectively, while the prediction for Deception, which
has a complex topography, has an RMSE of 56.2 m. Further details on the predictions of the observable ice sheet features can
be found in Appendix A.

## 4 Discussion

We first discuss the effect of informing PINNs with different physical constraints for the Deception region. After determining
the most effective physical constraints for inferring the bed topography, we discuss PINN results for the three regions of
interest. While the focus of this paper is on PINN predictions of the ice thickness, and subsequently the bed topography, a brief
discussion of the PINN predictions for the second unknown, the basal friction, can be found in Appendix B.



## 4.1 Determination of Physical Constraints

From the results presented in Fig. 4, we observe that all three PINNs (i.e., MB, SB, and MB+SB) predict significantly different bed topographies than that of BedMachine, especially outside $\Omega_{mc}$. All three PINNs satisfy ground-truth bed topography data better than BedMachine outside $\Omega_{mc}$, as indicated by the RMSEs in Table 2. However, PINN (MB) and PINN (SB) predict

isolated, crater-like features along radar flight tracks as shown in Fig. 4(a,b). Given the presence of fast-flowing, tributary ice streams in the Deception region, these crater-like bed features are unrealistic. We expect that PINN (MB) and PINN (SB) are fitting the training data too closely and not interpolating or 'connecting' inferred bed features. In other words, PINN (MB) and PINN (SB) are overfitting the training data, resulting in artificially lower RMSEs than that of BedMachine outside $\Omega_{mc}$. PINN (MB+SB), however, predicts several troughs that coincide with the fast-flowing ice streams and satisfy ground-truth

bed topography data outside of $\Omega_{mc}$. Therefore we observe that PINN (MB+SB) successfully connects inferred bed features, leading to a far more realistic bed topography map for Deception. The PINN (MB+SB) RMSE of 150 m outside $\Omega_{mc}$ is slightly higher than the PINN (MB) and PINN (SB) RMSEs of 133 m and 108 m respectively. This modest increase in the RMSE is expected, as PINN (MB+SB) uses multiple physical constraints, introducing additional regularization terms in its loss function. As a result, the slightly higher RMSE indicates that PINN (MB+SB) is learning a more generalized solution for

the whole Deception region rather than overfitting the training data.

Within $\Omega_{mc}$, the difference between BedMachine and PINN (SB) in Fig. 4(e) is more pronounced. Yet, interestingly, the PINN (SB) RMSE of 71.3 m is similar to the BedMachine RMSE of 76.2 m within $\Omega_{mc}$ (see Table 2). The differences of over 500 m in Fig. 4(e) could be explained by the different physical constraints imposed within $\Omega_{mc}$ (i.e., PINN (SB) imposes stress balance while BedMachine imposes mass balance), though a more likely explanation for the PINN (SB) RMSE being low is due

to overfitting. On the other hand, the reduced difference within $\Omega_{mc}$ for PINN (MB) and PINN (MB+SB) confirms that these PINNs are indeed using mass conservation to inform predictions. Yet, both PINN (MB) and PINN (MB+SB) predict thinner ice than that suggested by BedMachine and ice thickness measurements, failing to capture the full depth of the troughs beneath Deception Ø Center North, Deception Ø Center South, and Uunartit Islands. Indeed, the PINN (MB) and PINN (MB+SB) RMSEs of 98.5 m and 105 m respectively is higher than the BedMachine RMSE even though the physical constraints imposed

for all three are the same. We attribute this difference to the fact that BedMachine solves for the ice thickness using observations of ice velocity and the apparent mass balance, while the PINNs make predictions of these observable ice sheet features that are then substituted into the mass balance residual (in the loss function). Figure A1 for PINN (MB+SB) depicts that there are likely to be differences between predicted and ground-truth ice velocities, likely leading to uncertainties in the computation of the mass balance residual. Furthermore, training the PINNs involves minimizing the loss function (instead of directly solving

for ice thickness). These reasons imply that BedMachine imposes mass conservation more 'strongly' within $\Omega_{mc}$, while the PINNs impose the physical constraints more 'weakly' (i.e., as physical loss terms only). This may explain why BedMachine is able to resolve the full depth of the troughs beneath Deception Ø Center North, Deception Ø Center South, and Uunartit Islands within $\Omega_{mc}$ better than the PINNs.





To summarize, our results indicate that constraining the PINN with mass balance alone and stress balance alone is not suf-
ficient for predicting realistic bed topography in complex regions, like Deception. Both PINN (MB) and PINN (SB) are prone
to overfitting, especially outside $\Omega_{\mathrm{mc}}$. We determine that PINN (MB+SB) is most effective for inferring the bed topography as
multiple physical constraints help mitigate overfitting, allowing for a more generalized prediction of the bed topography. This
PINN (MB+SB) framework draws on the strengths of multiple physical constraints and exceeds their limitations while also
fitting the training data, allowing for more realistic predictions.

**4.2   Applications to Regions of Interest**

We examine the predicted bed topography for Upernavik, shown in Fig. 5(a). The difference map shown in Fig. 5(d) reveals
that BedMachine and the PINN have broadly similar bed topography maps because they conserve the physical laws for the
entire Upernavik region. The PINN is successful at "connecting" and extrapolating ice thickness measurements, revealing a
realistically-looking, continuous trough beneath Nunatakassaap Sermia and an extended trough beneath the northern fork of
Upernavik Isstrøm North, both of which are not captured in BedMachine. Given that the PINN RMSE of 70.6 m is lower than
that of BedMachine, we expect that these new bed features satisfy ground-truth bed topography data and are realistic. Yet, in
some cases where the PINN has no data points to further constrain it's predictions, it may not capture important bed features.
For instance, the PINN does not infer a continuous trough beneath the southern fork of Upernavik Isstrøm North where there
are no ice-penetrating radar measurements (see the ice thickness measurements along flight tracks in Fig. 2(d)). This means
that the PINN would need to rely completely on the mass balance and stress balance residuals to infer the ice thickness, and
subsequently the trough beneath. When we have no data points within $\Omega_{\mathrm{mc}}$ to constrain predictions, we expect that BedMachine
will produce more realistic results as it imposes mass conservation more 'strongly' within $\Omega_{\mathrm{mc}}$, while the PINN enforces mass
conservation and momentum conservation 'weakly' (as discussed in Section 4.1). This may explain why the PINN did not
successfully capture a continuous trough beneath the southern fork of Upernavik Isstrøm North, where the PINN has no data
points to further constrain its predictions of ice thickness.

For Narssap, we note that $\Omega_{\mathrm{mc}}$ generally encompasses the region with ice velocities that are $\sim 100$ m y$^{-1}$ or greater (see
Fig. 2(e)). The PINN suggests glacial valleys outside $\Omega_{\mathrm{mc}}$ which are not captured in BedMachine and satisfy ground-truth
ice-penetrating radar measurements (see RMSEs in Table 3). We find these bed features to be realistic as they coincide directly
with faster-flowing sectors, where velocities are approximately 50-100 m y$^{-1}$. Within $\Omega_{\mathrm{mc}}$, we find that the PINN RMSE is
lower than the BedMachine RMSE, implying that the PINN is satisfying ground-truth measurements better than BedMachine.
However, in the fast-flowing region, we note that the PINN predicts thinner ice than that shown in Fig. 2(h) and Fig. 3(b) beneath
Narssap Sermia and Akullersuup Sermia, implying that it is not able to recover the depth of the trough beneath these outlet
glaciers. As mentioned above, PINN predictions of the ice velocity have larger uncertainties for the fast-flowing regions, such
as Narssap Sermia and Akullersuup Sermia, which likely affects the computation of the PDE residuals. The PINN predictions
of the surface elevation are smooth, which could also contribute to uncertainty in the PDE residuals, but since the PINN RMSE
of 12.3 m for the surface elevation is relatively small we expect that most of the uncertainty in the PDE residuals is due to the
velocity predictions. Furthermore, we observe that Narssap is a region with sparse ice thickness measurements and that the





PINN conserves mass and momentum more 'weakly' than BedMachine which 'strongly' imposes mass balance along these fast-flowing regions within $\Omega_{mc}$, likely explaining why BedMachine is able to recover the depth of the trough beneath Narssap

Sermia and Akullersuup Sermia, while the PINN is not as successful.

The Deception region is known to have a complex topography and we note that $\Omega_{mc}$ is quite small, covering only the margins of the region as indicated by the black outline in Fig. 5(f), where the ice velocities are approximately $1000 \text{ m y}^{-1}$ or greater. The PINN unveils an intricate network of troughs outside $\Omega_{mc}$ that satisfy ground-truth bed topography measurements with lower RMSEs than BedMachine (see Table 3). Similar to Narssap, we find that these inferred features coincide with fast-flowing

ice streams outside $\Omega_{mc}$. Yet, also similar to Narssap, the PINN predicts thinner ice than shown in Fig. 2(i) within $\Omega_{mc}$ and consequently does not quite capture the full depth of the trough beneath Deception Center North, Deception Center South and Uunartit Islands. Similar to previous cases, this difference is explained by the fact that $\Omega_{mc}$ encompasses the region with fast-flowing ice where the PINN predictions of the ice velocity are more uncertain (see Fig. A1(f)) and by the fact that the PINN imposes the conservation laws weakly compared to BedMachine.

**4.3   Limitations and Next Steps**

Regarding the differences in the PINN predictions of the state variables (i.e., velocity, apparent mass balance, and surface elevation), we expect that these can be explained by the PINN model resolution. In other words, these differences are likely the result of too few data points provided to the PINN during the training process. We observe that the PINN better captures observable features with gradual, smoother transitions (or gradients) as opposed to those with sharper transitions. For instance,

the PINN accurately predicts the apparent mass balance which changes far more gradually across the regions of interest, but struggles to capture faster flowing ice velocities that have sharper gradients. As a result, increasing the number of data points and providing the PINN with more spatial information regarding these observable features will help limit the differences in all state variables predictions. That being said, it should be noted that increasing the number of data points and collocation points would lead to an increase in computational cost. Furthermore, we expect that we will not be able to limit small differences

altogether with our current framework due to the nature of the PINN architecture itself (Krishnapriyan et al., 2021); the denoising nature of the PINN lends itself to smoother, lower-frequency predictions. To address this limitation, we may have to employ a multi-stage training scheme (Wang and Lai, 2024).

**5   Conclusions**

In this paper, we use PINNs to solve a two-dimensional inverse problem, set up using the open source package, PINNICLE.

With this framework, we improve inference of ice thickness, and subsequently the bed topography beneath the GrIS. We apply our framework to three diverse regions: Upernavik, Narssap, and Deception, to assess its performance in different glaciological settings. By testing PINNs constrained by mass conservation alone, momentum conservation alone, and both conservation laws, we determine that using both mass balance and stress balance is most effective for slower-moving sectors with sparse measurements and complex bed topographies. Our method is especially effective outside the region where BedMachine uses a



mass-conserving approach, discovering new bed features beneath Narssap and Deception. We conclude that the PINN frame-
work provides a viable alternative to BedMachine and is especially well-suited to inferring bed features where BedMachine
does not use a mass-conserving approach. We recommend to use this approach to infer the bed topography in the ice sheet
interior where BedMachine relies on simple interpolation methods and where PINNs would provide a more physically based
estimate of the bed topography, especially when constrained with two conservation laws.

*Code and data availability.*  The Python and MATLAB scripts, pre-processed data files, saved PINN weights, parameter files, and training
histories for each of the experiments described in this study are hosted on GitHub at https://github.com/mansakrishna23/BedMappingPINN.
git and archived on Zenodo at https://doi.org/10.5281/zenodo.16852920. The PINNICLE source code and development history are hosted on
GitHub at https://github.com/ISSMteam/PINNICLE. The specific version of PINNICLE used in this study has been archived on Zenodo and
is available at: https://doi.org/10.5281/zenodo.14889235 (Cheng et al., 2025a). The PINNICLE software is licensed under the GNU Lesser
General Public License v2 (LGPLv2). The Ice-sheet and Sea-level System Model (ISSM, Larour et al., 2012) source code is open-source
and available at https://doi.org/10.5281/zenodo.7850841. The PINN training data are retrieved from the following sources: ice velocity data
are retrieved from NASA MEaSUREs products (Joughin et al., 2018), the apparent mass balance data are retrieved from RACMO 2.3 (Noël
et al., 2016) and ICESat-2 (Smith et al., 2020), the surface elevation training data are from the Greenland Ice Mapping Project (GIMP, Howat
et al., 2014), and the ice thickness data are retrieved from ice-penetrating radar measurements from CReSIS Radar Depth Sounder Data
(CReSIS, 2016).



## Appendix A: Extended Results

Using the PINN (MB+SB), we predict the ice velocities for each region, the magnitudes of which are shown in Fig. A1(a-c). To evaluate how well this PINN captures ice velocities, we calculate the difference between the velocity training data from NASA MEaSUREs (Joughin et al., 2018) and the predicted ice velocities; these difference maps are shown in Fig. A1(d-f). The difference maps show that the PINN is generally able to capture the velocities of both fast-flowing and slower-moving ice (RMSEs are shown in Table 4), though predictions are less accurate along fast-flowing ice streams. Upernavik and Narssap velocity predictions have lower RMSEs of 40.7 m y$^{-1}$ and 24.8 m y$^{-1}$ respectively, while the Deception velocity prediction has a higher RMSE of 62.8 m y$^{-1}$.

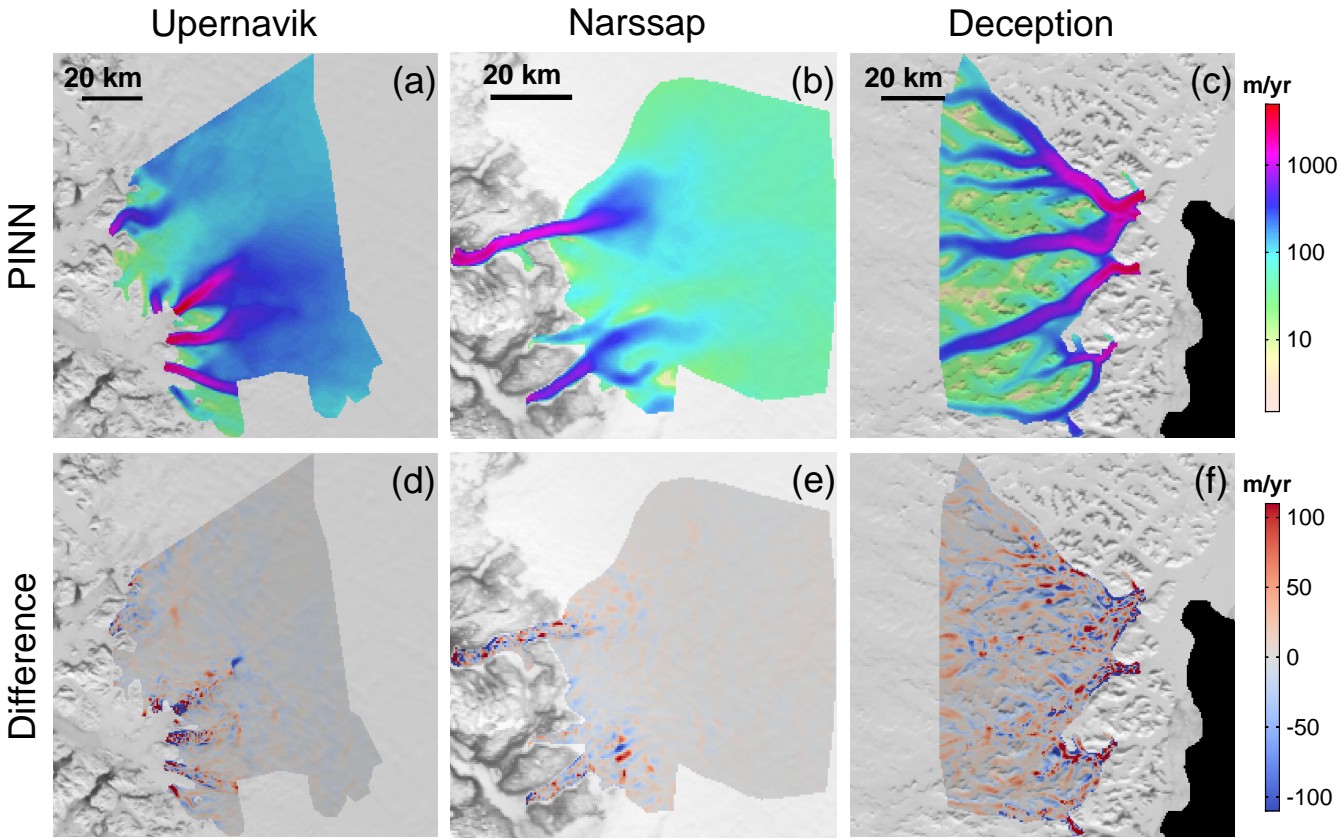

**Figure A1.** PINN (MB+SB) predicted ice velocity and difference maps for (a,d) Upernavik, (b,e) Narssap, and (c, f) Deception. (a-c) depict the PINN (MB+SB) predicted ice velocity maps and (d-f) depict the difference between ice velocity from NASA MEaSUREs shown in Fig. 2(a,e,i) and PINN velocity maps.

Using the PINN (MB+SB), we predict the apparent mass balance for each region, shown in Fig. A2(a-c). To evaluate how closely the PINN captures the apparent mass balance, we calculate the difference between the training data from RACMO 2.3 (Noël et al., 2016) and ICESat-2 derived ice thinning rates (Smith et al., 2020) and the predicted apparent mass balance; these



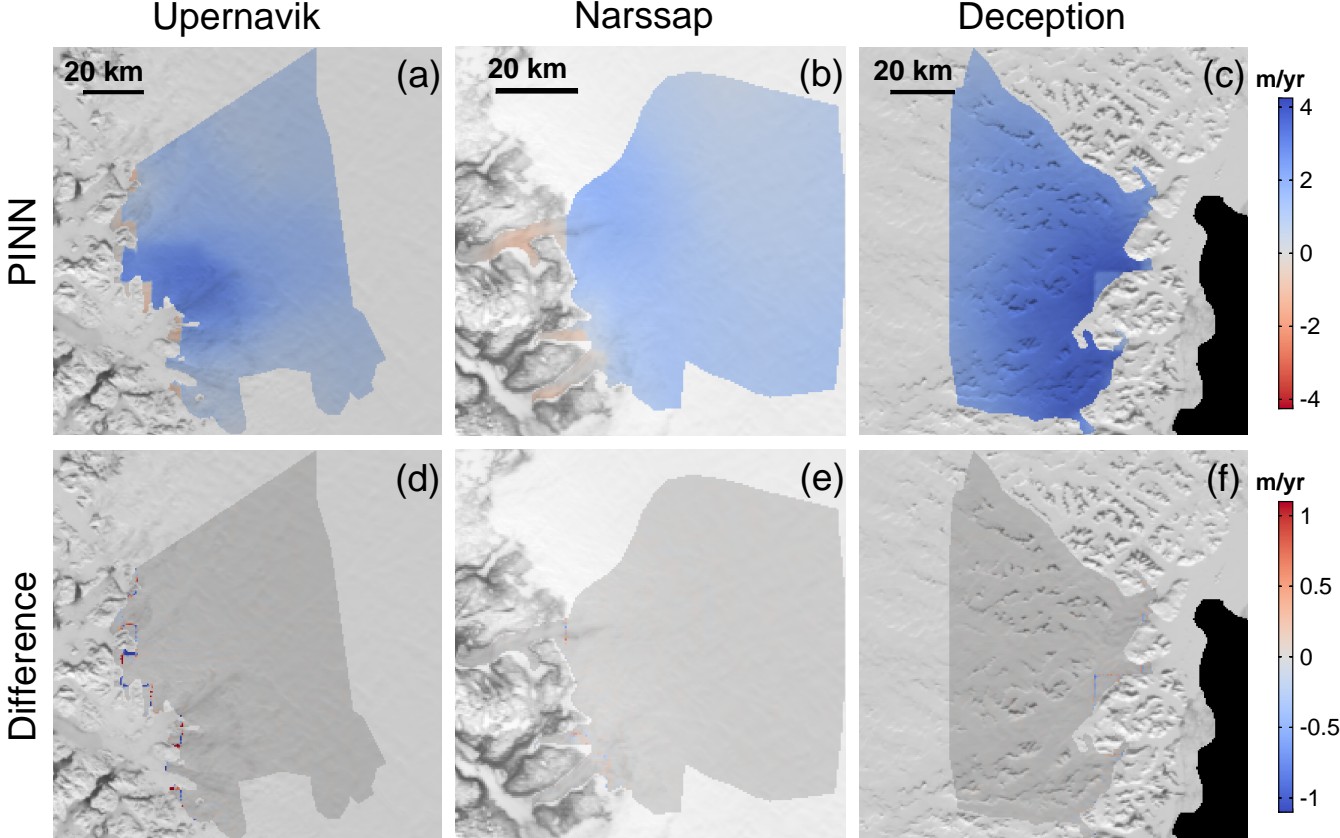

**Figure A2.** PINN (MB+SB) predicted apparent mass balance and difference maps for (a,d) Upernavik, (b,e) Narssap, and (c, f) Deception. (a-c) depict the PINN (MB+SB) predicted apparent mass balance maps and (d-f) depict the difference between apparent mass balance from RACMO 2.3 and ICESat-2 shown in Fig. 2(b,f,j) and PINN apparent mass balance maps.

difference maps are shown in Fig. A2(d-f). The difference maps show that the PINN successfully captures the apparent mass balance (RMSEs shown in Table 4) within $\pm 0.5$ m y$^{-1}$ for all three regions. Narssap and Deception apparent mass balance predictions have lower RMSEs of 0.0209 m y$^{-1}$ and 0.0321 m y$^{-1}$ respectively, while the Upernavik apparent mass balance prediction has an RMSE that is an order of magnitude higher at 0.112 m y$^{-1}$.


Using the PINN (MB+SB), we predict the surface elevation for each region, shown in Fig. A3(a-c). To evaluate how well the PINN predicts the surface elevation, we calculate the difference between the surface elevation training data from GIMP (Howat et al., 2014) and the predicted surface elevation, shown in Fig. A3(d-f). The difference maps show that the PINN does generally capture the surface elevation (RMSEs are shown in Table 4), though these predictions are far smoother than observations suggest. Upernavik and Narssap surface elevation predictions have lower RMSEs of 20.9 m and 12.3 m respectively, while the Deception surface elevation prediction has a higher RMSE of 56.2 m.






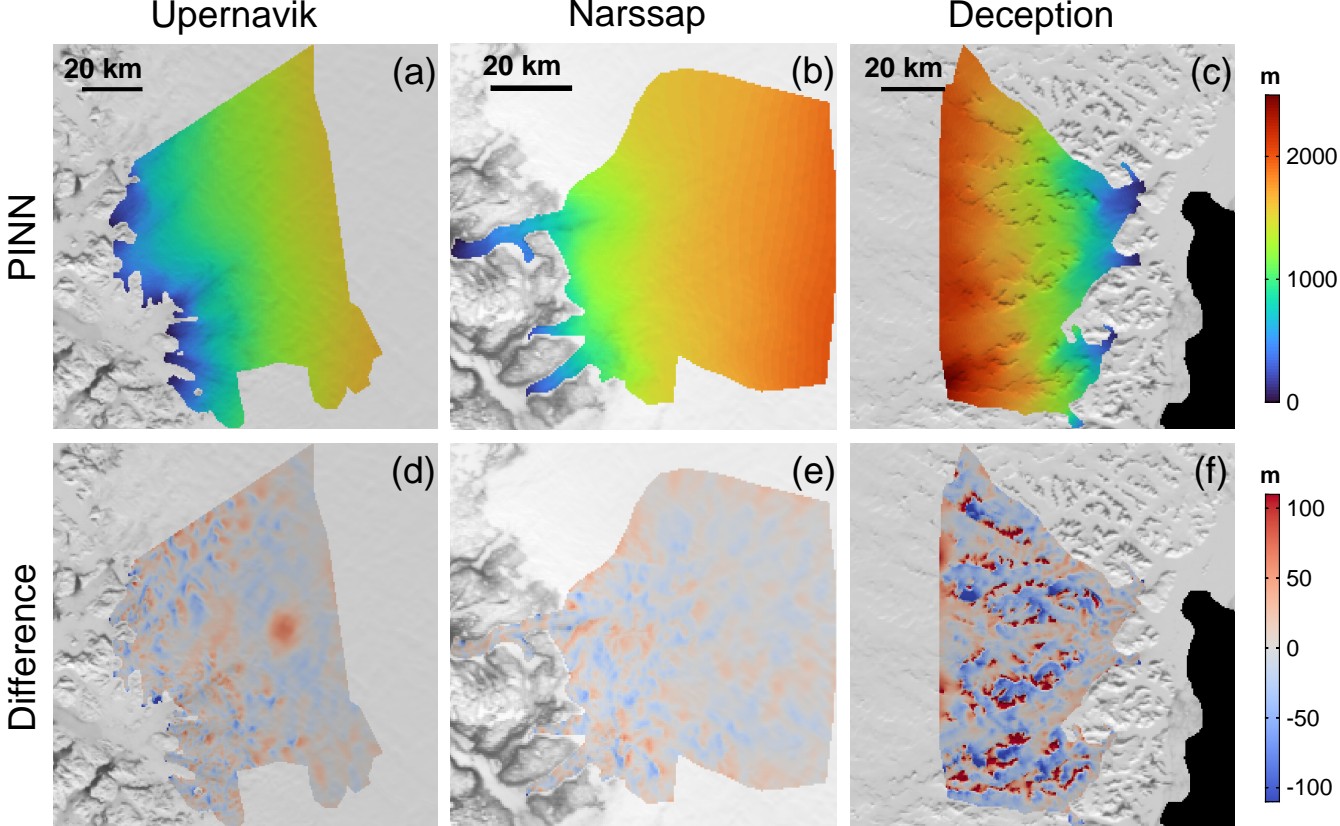

**Figure A3.** PINN (MB+SB) predicted surface elevation and difference maps for (a,d) Upernavik, (b,e) Narssap, and (c, f) Deception. (a-c) depict the PINN (MB+SB) predicted surface elevation maps and (d-f) depict the difference between surface elevation from GIMP shown in Fig. 2(c,g,k) and PINN surface elevation maps.

Lastly, using the PINN (MB+SB), we predict the basal friction coefficients for each region. To assess the quality of the PINN predictions of the basal friction coefficient, we invert for the basal friction coefficients using the Ice-sheet and Sea-level System Model (ISSM, Larour et al., 2012) with the method described in Morlighem et al. (2013). Note that to obtain the

inverted basal friction coefficients, we initialize our ISSM models with BedMachine ice thickness values. Given the formulation of Weertman's friction law, Eq. (5), we plot maps of the PINN predicted squared basal friction coefficient values ($C^2$) that are used to calculate the basal shear stress; these are shown in Fig. A4(b,e,h). Figure A4(a,d,g) depict the squared basal friction coefficient values ($C^2_{\mathrm{ISSM}}$) obtained through an inversion using ISSM. The differences between these two squared basal friction coefficient maps are shown in Fig. A4(c,f,i).

We observe that the inverted and predicted squared basal friction coefficients for Upernavik and Narssap share some similarities, with lower coefficients in regions of fast-flowing ice and higher coefficients in regions where the ice is slower-moving. However, we also observe these maps have notable differences of $\sim 1 \times 10^7$ Pa s$^3$ m$^{-3}$ located in the areas of fast-flowing ice.





**Figure A4.** Squared basal friction coefficients and difference maps for (a-c) Upernavik, (d-f) Narssap, and (g-i) Deception. (a,d,g) depict the ISSM inverted squared basal friction coefficients, (b,e,h) depict the PINN (MB+SB) predicted squared basal friction coefficients, and (c,f,i) depict the difference between the inverted and predicted squared basal friction coefficients.

For Deception, we observe that the squared basal friction coefficient maps are significantly different, especially in the region where the PINN predicts an intricate network of troughs, outside the MC domain, where the PINN bed topography is over 600
m deeper than that of BedMachine.



**Appendix B: Extended Discussion**

While the focus of this paper is on PINN predictions of the ice thickness, and subsequently the bed topography, we briefly discuss PINN (MB+SB) predictions of our second unknown: the basal friction coefficient. From Fig. A4, we observe that though the inverted and predicted basal friction coefficients share some similarities, there are significant differences that likely

occur for a few reasons. First, since the PINN has no training data with which to constrain its predictions for the basal friction coefficient, it must rely solely on the PDE constraints. Yet, as established earlier, there are often uncertainties associated with the PINN predictions of observable ice sheet features that are substituted into the PDE residuals, especially along fast-flowing ice streams. Second, we expect that some of the difference in the basal friction coefficients can be explained by the PINN model resolution. Since the PINN is relying solely on the PDE constraints to infer the basal friction coefficients, it will only

compute these for the randomly selected collocation points within a region of interest. Therefore, we expect that increasing the number of collocation points (i.e., increasing the PINN model resolution) could improve PINN predictions of the basal friction coefficients. Lastly, minor differences could be explained by the differences in model geometries as the inverted basal friction coefficients rely on BedMachine ice thickness while the PINN basal friction coefficients use the PINN predicted ice thickness. Indeed, we observe that there is a significant difference in bed topography of over $\sim 600$ m in the northwest sector

of the Deception region, where we also observe that the difference in basal friction coefficients is more pronounced (see Fig. 5(f) and Fig. A4).



## Appendix C: Validation Study

We assess the accuracy of our PINN framework (informed with two conservation laws) by training a PINN for the Upernavik region shown in Fig. 2(a-d) and 3(a). We find this region to be appropriate for a validation study as the bed topography is well
constrained with dense ice thickness measurements as shown in Fig. 2(d).

We train the PINN using randomly selected data points from a subset of the available ice thickness measurements along ice-penetrating radar flight tracks and keep the rest of the ice thickness measurements 'hidden', which will be used later for testing the accuracy of PINN predictions. Most notably, we hide the majority of the ice thickness measurements along Upernavik Isstrøm North to ascertain whether the PINN can infer the trough beneath from sparse ice thickness observations. Henceforth,
we refer to the PINN trained for this validation study as the "PINN (V)" (or the Validation PINN).

We compare PINN (V) predictions of the bed topography to ground-truth bed topography data, obtained along ice-penetrating radar flight tracks, $\hat{b}_{\mathrm{data}}$. Despite being exposed to randomly sampled ice thickness measurements from the training subset (black lines only) of flight tracks in Fig. C1(c), we find that PINN (V) predicts a bed topography map similar to BedMachine, with differences less than ∼250 m. Most importantly, the PINN (V) is able to recover a trough along Upernavik Isstrøm North,
despite being exposed to limited ice thickness measurements.

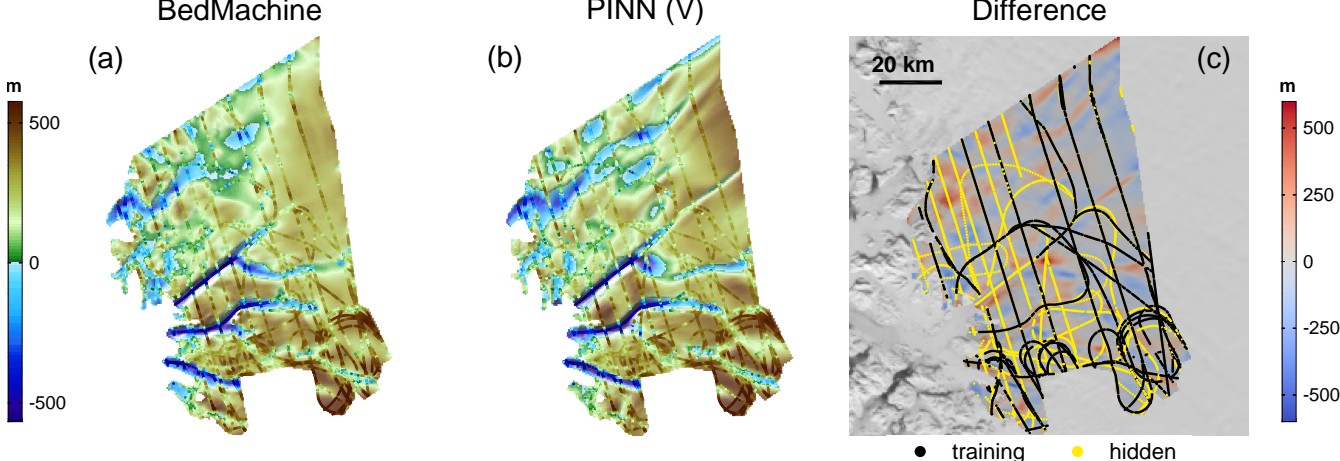

**Figure C1.** Inferred bed topography maps for Upernavik from BedMachine and PINN (V). (a,b) shows BedMachine and PINN (V) predicted bed topography maps respectively, which we overlay with ground-truth bed topography data calculated along ice-penetrating radar flight tracks. (c) depicts the difference between the BedMachine and PINN (V) bed topography maps. We also plot the subset of flight tracks from which we select data points for training the PINN (V) (i.e., the black lines) and the subset of flight tracks that are hidden for validation (i.e., the yellow lines) on the difference map.

We further quantify the error in PINN (V) predictions, showing the RMSE between ground-truth and predicted bed topography values along ice-penetrating radar flight tracks in Table C1. For comparison we also include the RMSEs between ground-truth and BedMachine bed topography values in Table C1. Along the training flight tracks (see Fig. C1), to which both



BedMachine and PINN (V) are exposed, the PINN (V) RMSE of 64.4 m is lower than the BedMachine RMSE of 88.3 m,
implying that PINN (V) is satisfying the ground-truth observations better than BedMachine. However, along the hidden flight
tracks we observe that the PINN (V) RMSE of 114 m is slightly higher than the BedMachine RMSE of 97.9 m. This is likely
because BedMachine has been exposed to the ice-penetrating radar measurements along the hidden flight tracks, whereas PINN
(V) has been exposed to none of these measurements. Given that BedMachine has been exposed to all the hidden flight tracks
and the PINN (V) has been exposed to none of them, we find it highly promising that these RMSEs are similar. Thus, the simi-
lar bed topography maps in Figures C1(a,b) and the similar RMSEs of BedMachine and PINN (V) show that by coupling mass
balance with stress balance in our PINN framework, we are indeed able to predict a realistic bed topography using observable
ice sheet features and sparse ice thickness measurements.

| Flight Tracks | PINN vs. $b_{\text{data}}$ (m) | BedMachine vs. $b_{\text{data}}$ (m) |
|---|---|---|
| Training | 64.4 | 88.3 |
| Hidden | 114 | 97.9 |

**Table C1.** RMSEs for the validation study on Upernavik along 'training' and 'hidden' ice-penetrating radar flight tracks.

*Author contributions.* MK, GC, and MM designed this study. MK performed the numerical computations. MK wrote the manuscript with
input from GC and MM.

*Competing interests.* GC is a member of the editorial board of the The Cryosphere. All other authors declare that they have no conflict of
interest.

*Acknowledgements.* MK, GC, and MM are grateful for the assistance of Sade Francis in training preliminary versions of the PINN models.
MK, GC, and MM acknowledge support from the National Science Foundation (award no. 2118285), "HDR Institute: HARP- Harnessing
Data and Model Revolution 535 in the Polar Regions." MK acknowledge the Texas Advanced Computing Center (TACC) at The Univer-
sity of Texas at Austin for providing computational resources that have contributed to the research results reported within this paper. URL:
http://www.tacc.utexas.edu. In particular, MK would also like to acknowledge the support of Sikan Li who helped with using the compu-
tational resources at TACC. GC acknowledges support from the Novo Nordisk Foundation under the Challenge Programme 2023 (Grant
NNF23OC00807040). Lastly, MK, GC, and MM acknowledge the use of data and/or data products from CReSIS generated with support
from the University of Kansas, NSF grant ANT-0424589, and NASA Operation IceBridge grant NNX16AH54G.



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
