# Peer review of "Inferring subglacial topography using physics informed machine learning constrained by two conservation laws"

_EGUsphere, 2025_

## Author Comment (AC1)

**Response to the Reviewers**

January 4, 2026

**Response to Reviewer 1**

**Summary and general comments**

This is a well-organized and well written paper describing the application of physics-informed neural networks (PINNS) for further improving the generation of sub-glacial bed topography datasets, building on previous "BedMachine" efforts that have been ongoing for the past decade or so. Two conservation equations – the continuity equation and a momentum balance equation for ice flow – are considered and introduced as additional constraints on the loss function, akin to their introduction as constraints in the "cost function" of PDE-constrained optimization (a good analogy to consider including for readers more familiar with the language of glaciology modeling / optimization?). The two constraints are considered on their own and in combination and the resulting bed topography datasets are compared and contrasted with previous BedMachine results for three glaciologically distinct regions. Overall, the authors argue convincingly that the new approach has merit and demonstrates potential for improving inferred bed topography in regions where the traditional BedMachine approach begins to break down. Overall, the work seems very worthy of publication and readers of The Cryosphere will find it a worthy contribution. My suggestion would be to accept for publication with minor revisions, noting that these are those suggested revisions (detailed below and identified by their line number in the submitted version) are largely editorial in nature.
**Response:** We appreciate the reviewer's thorough and thoughtful comments. Thank you for your encouraging feedback and for emphasizing the overall quality and relevance of our work.

My one more substantial suggestion – not necessarily for this publication but possibly for a future effort – is that I think it would be very useful to redo this exercise for a single region (I realize the computational cost could be a challenge, so pick a single region, like the most challenging one discussed here) but using L1L2 (Blatter/Pattyn) for the momentum balance model as opposed to SSA (which you've already done and could reuse as the baseline). Because the former allows for internal deformation, a possibly very different vertical velocity profile (and hence depth-averaged velocity and flux divergence) might be implied in regions of slower moving ice (noting that the modeled 2d surface velocity field could still be used as the velocity constraint, so that there should not necessarily be any significant reformulation of the loss functions discussed herein). It would be interesting to see if the more accurate stress balance constraint helped to alleviate any of the remaining problems discussed below.
**Response:** We appreciate the reviewer's feedback about using L1L2 (Blatter/Pattyn) for the momentum balance model to account for vertical shear. While this is currently outside the scope of this paper, we will include this in our discussion of future work.

**Detailed comments**

21: Are these refs now considered the definitive source for defining the amount of potential SLR locked up in the ice sheets? If not, maybe consider adding one or two more from other authors for the sake of diversity?
**Response:** We believe these references: IMBIE Mass Balance Intercomparison (Otosaka et al., 2023) provides the latest consensus on the current contribution to sea level rise and BedMachine (Morlighem et al., 2017; Morlighem, 2020) and Bedmap 3 (Pritchard et al., 2025) are the most up to date estimates for defining

the amount of potential sea level rise. We will include these in the manuscript.

23: "numerical ice sheet modeling"
**Response:** Thank you for the suggestion, we will make this change in the manuscript.

25: It seems like a summary-level reference might also be appropriate here (?), e.g. something from one of the recent IPCC reports (that integrates results from a large number of individual publications).
**Response:** Thank you for the suggestion. In addition to Aschwanden et al. (2021), we will also add the ISMIP6 papers as reference for multi-model ensembles.

28-29: It should probably be noted here that these experiments were assuming a marine ice sheet with a significant over-deepening inland (since this configuration would necessarily be more sensitive than say, an ice sheet grounded above sea level).
**Response:** We will include a note that these experiments were assuming a marine ice sheet with a significant over-deepening inland, and therefore likely more sensitive.

37-38: Is the ~2 km limit proposed here coming from the Durand et al. (2011) paper? While I more-or-less agree with this idea, I don't know that this single reference is adequate to support the precision implied by this statement. Maybe consider softening it a little bit to something less precise, e.g. "order km-scale spatial resolution"?
**Response:** The sentence will be modified to state "kilometer-scale spatial resolutions" required by ice sheet models instead.

63: "three regions in Greenland"; maybe add a few words of clarification here that they are glaciologically distinct / different? E.g., presumably you mean regions where velocity occurs primarily via fast sliding, a region where it occurs via a mix of sliding and deformation, etc.
**Response:** The sentence will be modified to state we have chosen "three glaciologically distinct" regions.

Figure 1 caption: "The loss function is comprised of . . . " or "The loss function includes data loss . . . "
**Response:** Caption will be changed to state "the loss function is comprised of...".

89: "fully connected layers", maybe use "fully connected ('dense') layers . . . " ?
**Response:** Thank you for the suggestion, we will make this change in the manuscript.

101: Should it be "the apparent mass balance residual" ?
**Response:** The "mass balance residual" in this case refers to the residual of the conservation of mass. The "apparent mass balance" is defined as $\dot{a} = \dot{M}_s - \dot{M}_b - \partial H / \partial t$. To minimize confusion, we will remove the term "apparent mass balance" from the manuscript and re-phrase our explanation.

103-105: I am guessing that maybe this is discussed further below (?), but it seems like you are already potentially limiting the usefulness of this approach by restricting the momentum balance to SSA. I.e., if one of the main interests here is in improving the inference in regions of slower moving ice flow, which is presumably due to less sliding and more internal deformation, then SSA doesn't seem like the right assumption to make for the model dynamics. I know that ISSM has higher-order approximations available (e.g., L1L2 or "Blatter-Pattyn"). Has that also been explored (acknowledging the obvious additional computational burden) and compared against the approach using SSA?
**Response:** We decided to start with a simpler approximation of the momentum balance (i.e., SSA) to minimze computational cost and to see if this method of using two conservation laws could produce sensible results. Further, since we wanted our method to be comparable to the method in BedMachine, we needed to use the depth-averaged conservation of mass and so also needed to use a 2D approximation of the momentum balance. That being said, for the interior, slower-moving regions of the ice sheet we hope to use higher-order approximations of the momentum balance, and we will include this as an additional point in our discussion section.

116: "... from THE regional climate model RACMO..."
**Response:** Thank you for the suggestion, we will make this change in the manuscript.

Section 2.3: It sounds like the basal mass balance term in equation 2 is assumed to be 0? If so, it would be good to note that explicitly here in the discussion of the apparent mass balance term.
**Response:** We will make this change and explicitly mention that the basal mass balance term is assumed to be negligible.

152: "to prevent from taking" (omit "from"?); "or diving by zero" ("dividing by zero")
**Response:** Thank you for the suggestion, we will make this change in the manuscript.

176-177: Would it be worth commenting on the choice of median vs. mean? Is the median chosen because of the small number (5) of samples, such that the mean could be easily biased?
**Response:** Yes, we chose to use the median because since we are using an ensemble of five models for each region, and the mean could easily be biased. We will mention this explicitly in the methods section.

180: By "challenging to implement", do you mean where the traditional / previous mass conservation approach does not perform well? Implementation sounds more like the approach is challenging, but I imagine the approach is just as easy to implement in these regions, it's more the prior / baseline result that you are not happy with.
**Response:** By "challenging to implement", we do mean that the mass conservation inverse approach does not perform well since the ice is moving slowly. We will make this more explicit within the manuscript.

242: It's not clear here exactly what "Fig.2(1)" is referring to.
**Response:** Fig. 2(l) refers to the ice thickness values along ice-penetrating radar flight tracks for the Deception region. To minimize confusion, the sentence will be modified to read "ice-penetrating radar flight tracks in Fig. 2(l)".

Figure 5: In the caption for this and figure 4 it would be helpful to remind the reader which dataset is subtracted from which and shown in panels d-f (e.g., PINN minus original BedMachine product or vice versa?).
**Response:** We will definitely make this more explicit in the Fig. 4 and Fig. 5 captions. We will state that the difference is $\hat{\boldsymbol{b}}_{\mathrm{BedMachine}} - \hat{\boldsymbol{b}}$ for panels d-f.

Table 4: It might also be useful to provide some percent / fractional metrics here? E.g., for the apparent mass balance RMSE, how does that number compare to the average apparent mass balance over the same area? Such a table could be added to the SI if it's not deemed important enough for the main text.
**Response:** We will include an additional column in Table 4 for the fractional/percent metrics.

3.2.2. – It's left hanging a bit as to the significance of the differences in u, apparent mass balance, and sfc. elevation when using the different approaches. For example, how do these differences compare to those that arise when using the original BedMachine approach? Would it make sense to include those metrics (differences in u, apparent mass balance, and sfc elevation) somewhere here for comparison? It's a bit unclear to me what the broader implications are of these secondary metrics w.r.t. using the derived datasets for modeling. If the authors have additional thoughts on this they would be welcome in the supplementary information.
**Response:** For the mass-conserving approach in BedMachine, the raw data (i.e., $\boldsymbol{v}_{\mathrm{data}}, \dot{a}_{\mathrm{data}}$) are used within the mass balance equation to directly invert for the ice thickness. With the PINN approach, in order to infer the ice thickness, the PINN must predict the $\boldsymbol{v}, \dot{a}, s, C$ fields that are used within the mass balance and momentum balance residuals. Therefore, the differences in these derived datasets arise due to the PINN architecture – while the PINN can minimize the error in these predicted fields with respect to the ground-truth data, it will never be exactly the same. We tried to explain this in the discussion section as BedMachine imposing mass conservation more 'strongly' and the PINN imposing the conservation laws more 'weakly', however we will develop this point further within our discussion to avoid confusion.

326: If the discussion starting in 4.1 is intended to be specific to Deception, then maybe that should be noted earlier in this paragraph? Alternatively, if the discussion in 4.1 up to line 326 where Deception is mentioned is supposed to be generic, then perhaps line 326 should be something more like, "... a far more realistic bed topography map, particularly for Deception."
**Response:** The discussion is meant to be specific to the Deception region, we will make this more clear in the manuscript.

329-330: Would "...slightly higher RMSE SUGGESTS ..." be more appropriate here than "indicates"? I think the speculation in this sentence makes sense, but it seems like it is perhaps speculation as opposed to a concrete fact.
**Response:** Thank you for the suggestion, we will make this change in the manuscript.

336-348: W.r.t. the prediction of thinner ice – is it also possible that this could be the result of the chosen stress balance model? E.g., in order for the SSA model to match surface velocities, it would need to assume a depth-averaged velocity profile that is larger than would be assumed in a model that allowed for internal deformation (E.g., L1L2), because SSA can only accommodate velocity via a change in the sliding component (unless I'm misunderstanding the model used here). If that is indeed the case, then it seems like the optimization process might necessarily bias the ice thickness on the thin side; if the depth averaged velocity is too large, the same flux (constrained by continuity equation and the apparent mass balance terms) can only be accommodated by reducing the ice thickness.
**Response:** This is a good point, however the region where the PINN predicts thinner ice is in the fast-flowing region where we might expect there to be more sliding rather than internal deformation (which is perhaps more typical in the slower-moving regions). While we will add this as a note to our discussion, we do think that the prediction of thinner ice in these areas is largely because the PINN 'struggles' to predict sharp transitions in the ice velocity.

345: "These reasons imply that ..." is a bit awkward. "This implies that ..."? "These arguments imply that ..." ?
**Response:** Thank you for the suggestion, we will make this change in the manuscript.

353: "... and exceeds their INDIVIDUAL limitations ..." ?
**Response:** Thank you for the suggestion, we will make this change in the manuscript.

398: "We observe that the PINN better captures..." → "We observe that the PINN captures observable features better with ..."
**Response:** Thank you for the suggestion, we will make this change in the manuscript.

403: "... state variable predictions". (remove plural on "variable")
**Response:** Thank you for the suggestion, we will make this change in the manuscript.

415: "... mass-conserving approach, AS CONFIRMED BY THE DISCOVERY OF new bed features beneath Narssap and Deception."
**Response:** Thank you for the suggestion, we will make this change in the manuscript.

417: "... we recommend USING this approach ..."
**Response:** Thank you for the suggestion, we will make this change in the manuscript.

A last thought / general comment: The implied "geomorphology" of the three focus areas studied here look very different from one another. E.g., The Upernavik and Narsaap beds look very smooth when compared to Deception. In the areas where there are no troughs, they almost look like high-resolution DEMs from past, heavily glaciated regions of Canada. Is there any published work on previous Greenland glaciations that might provide some more insight into this? I'm not suggesting it should be part of this paper, but it could be interesting to look into whether or not the "smoothness" that your methods are implying about the bed in different regions is in line with current glacial geological / geomorphological understanding. It

would seemingly be a further testament to the power of the methods used here if you were resolving that level of information about the bed through hundreds / thousands of meters of ice.

**Response:** Thank you for your comment. We agree that the geomorphology of the regions studied in this paper look quite different from each other. When we refer to "smooth" PINN predictions, we are comparing predictions directly with the data along ice penetrating radar flight tracks which we observe is far more "rough". Like BedMachine, the PINN also does capture lower-frequency approximations of the bed, however misses out on the higher-frequency details.

---

## Author Comment (AC2)

**Response to the Reviewers**

January 4, 2026

**Response to Reviewer 2**

This paper by Krishna et al. introduces a neural-network-based approach to infer bed topography constrained by both conservation of mass and momentum. The authors find that this approach can infer more physically realistic bed topography in slower-moving regions than BedMachine, which is only mass-conserving. The paper is well written and presents a valuable comparison between the two approaches. As PINN is a new method, such methodological studies are important for building an understanding of their benefits and current limitations. I have a few comments, summarized below:

**Response:** We appreciate the reviewer's thoughtful comments. Thank you for your positive feedback and for highlighting the relevance of our methodological study.

First, it's a good idea to hide part of the thickness data for independent validation in Appendix C. As $b = s - H$, the error reporting in the paper for $b$ is a direct result of the errors in $s$ and $H$, for which you have training data. My understanding is that the error reported in the main text between the prediction and the "ground truth" topography, along locations where $H$ and $s$ are provided to the NN, simply reflects how well the NN fits the $H$ and $s$ data. It is not relevant to the quality of the physics-based interpolation. A small error along the flight tracks does not imply good $b$ prediction between the tracks, so I'd be cautious about mixing the interpretation of data error in b with the success of the interpolation. That said, in Appendix C the prediction of $b$ at locations without $H$ data is the true demonstration of how effective the physics-informed interpolation is in regions where data are not directly available.

**Response:** Thank you for your comment, and for your positive feedback with respect to the validation study in Appendix C. Regarding the point about the uncertainty in the training data, we have indeed assumed that this data is the "ground-truth". We have tried to be careful about reporting the differences between predictions and the training data as "RMSEs" and have tried not to use the terms "error" or "uncertainty". However, to minimize confusion, we will explicitly state in table captions that the values reported are "differences" or "RMSEs" that should not be confused with "error" or "uncertainty". We will also explicitly discuss how the errors in the training data might affect the quality of our PINN predictions in the discussion section.

Second, the prediction of velocity from SSA is the depth-averaged velocity, but the observed velocity in the loss function is the surface velocity. Can the authors comment on when this distinction is important and why it is justifiable in the paper? Additionally, is SSA a good approximation of Stokes in all regions studied by the authors?

**Response:** It is indeed true that depth-averaged velocities are not always the same as surface velocities, which are used in the PINN loss function. Morlighem et al., (2011) which is one of the first references using a mass-conservation based approach for inferring glacial thickness accounts for this by incorporating a factor of 0.95 and a tolerance interval of $\pm 50$ m/yr for ice velocity within the optimization process (assuming that the depth-averaged velocities can be be 5 percent smaller than the surface velocity). Moreover, while the PINN loss function is set up to minimize differences between predicted velocities and the surface velocities, it also minimizes the mass balance and momentum balance residuals. As a result, the predicted velocity fields are unlikely to be identical to the surface velocity fields. Indeed, we have shown in Table 4 that the RMSEs of the velocity fields for each of the three regions are on the same order of magnitude as the tolerance interval in Morlighem et al. (2011), and consequently in BedMachine.

With regards to the use of SSA, several modeling studies, including models in the ISMIP6 ensemble (Goelzer

et al., 2020) use SSA to estimate the future sea level contribution of the Greenland Ice Sheet. That being said, it is also important to note that, in this case, we decided to start with a simpler approximation of the momentum balance to minimize computational cost and to see if this method of using two conservation laws could yield sensible results. Moreover, since we wanted our method to be comparable to BedMachine, we needed to use a 2D approximation of the momentum balance. For future work and for applying this method for the interior, slower-moving region of the ice sheet, we hope to use higher-order approximations of the momentum balance. We will elaborate on this in our discussion section.

Third, why does the mass-conserving PINN (MS) produce results so different from BedMachine (Fig. 4), if they both conserve mass? It is mentioned that PINN (MS) predicts isolated, unrealistic crater-like features along radar flight tracks due to "overfitting" (line 323), but why does the mass-conserving BedMachine not suffer from the same issue in the same region? It appears that PINN is using less data than BedMachine, but are they using the same thickness data? Can this difference in input data between PINN (MS) and BedMachine be made more explicit?

**Response:** The PINN (MB) and PINN (SB) do predict isolated bed features along the radar-flight tracks. However, these isolated features are located in the region *outside* of the mass-conserving domain of BedMachine, $\Omega \setminus \Omega_{\mathrm{mc}}$, as stated in the manuscript. In this area outside the mass-conserving domain, BedMachine uses other interpolation methods like kriging, whereas the PINN is attempting to constrain predictions to the conservation of mass and the conservation of momentum respectively. We have described this in our discussion section, but will work to make it clearer to minimize confusion.

With regards to the difference between the PINN training process and the BedMachine optimization process, we have mentioned that BedMachine uses all of the ice thickness training data whereas the PINN approach involves randomly selecting data points within the domain of interest in Appendix C. We have cited the BedMachine paper (Morlighem et al., 2017) where the details are well explained and documented, however, we will make this difference more explicit in our discussion.

Fourth, As PINN solves the equations weakly, the only measure of success, apart from data misfit, is the equation residual. In addition to the various data errors, I believe the paper needs to show map views of the equation residual to demonstrate convincing training success, and the PDE residual should be evaluated on a higher density of collocation points than the training collocation points to check whether the PDE is satisfied between the training collocation points.

**Response:** Thank you for the feedback. We will include these additional figures of the mass balance and momentum balance residuals in our manuscript.

Fifth, in the comparison of errors between different inversion results, it is only meaningful to say A is lower than B if the difference between them is larger than the uncertainties in A's errors. There are many discussions of error comparisons between PINNs and BedMachine, but the PINN errors are averaged over an ensemble of PINN predictions. It is important to consider the spread of errors among PINN predictions. In error reporting such as Table 2, I highly recommend including not only the error of the mean PINN prediction, but also error bars representing the range of error for each PINN prediction. Comparisons between errors are only meaningful after including the uncertainties in PINN errors due to the ensemble.

**Response:** We will also include figures that depict the standard error / range of the PINN ensemble for each region in our manuscript.

Finally, would it be possible for the authors to comment on the feasibility of enforcing both momentum and mass balance in the classical adjoint method? If this would be difficult, that would also strengthen the paper's narrative. This is a natural question that readers are likely to have and will be eager to hear the authors address.

**Response:** Using the classical adjoint method for this problem (i.e., in this case, attempting to invert for ice thickness and basal sliding coefficient using both conservation of mass and conservation of momentum) is technically difficult. The derivation with traditional adjoint method for a flow-line SSA problem can be found in Cheng et al. (2021), however, numerical implementation of 2D problem will require automatic differentiation. We will add this in the discussion to strengthen the paper's narrative.

**Minor comments**

Line 57-59: Literature review: Bolibar et al., 2023 did not use PINN. It is correct that Riel et al. (2021) is the first PINN study in glaciology. But Riel and Minchew (2023) appeared after some of the other cited studies, and thus I recommend moving it into the sentence along with other papers.
**Response:** We will make these changes in the manuscript.

Line 85: "satisfy the PDE residuals" → "satisfy the PDEs"
**Response:** Thank you for the suggestion, we will make this change in the manuscript.

Figure 1: Nice figure. Subscript "data" is missing in $L = L_{data} + L_\phi$
**Response:** Thank you for catching this typo! We will re-make the figure accordingly.

Line 86-87: Are your collocation points fixed in location or changing throughout iterations? Changing throughout iteration is highly recommended; if not doing so you'll likely overfit the physics on the fixed discrete collocation points.
**Response:** While it is possible to re-sample the collocation points during the training process and is a feature built-in to the PINNICLE package (Cheng et al., 2025), it can add to the computational cost because resampling often leads to slower convergence. We have chosen a high number of collocation points for this reason, so that the physical laws are generally conserved throughout the domain. Moreover, we also trained an ensemble of 5 PINN models with different random seeds. This resulted in the random selection of different collocation points for each of the PINNs, thereby reducing the likelihood of the PINN satisfying the physics for fixed collocation points. For each location, we then took the median prediction of the 5 PINNs. We will make this clear in the methods section. We will add a new test that allows for re-sampling of the collocation points for one of the regions. Based on our previous experiences during the development of PINNICLE (Cheng et al., 2025), we expect this result to be similar to the results of the experiments without resampling.

Line 115-117: As the apparent mass balance contains both ice thinning rate and the surface/basal mass balance, can you explicitly say how RACMO and ICESat-2 data are combined to give $\dot{a}$? Does ICESat-2 give a thinning rate $\partial H/\partial t$ without the effect of $\dot{M}_s, b$?
**Response:** Thank you for your feedback. We can make it more clear in the methods section that the surface mass balance $\dot{M}_s$ is from RACMO, the basal mass balance $\dot{M}_b$ is assumed to be negligible, and the ice thinning rates are derived from ICESat-2.

Table 1: Can you use the same time units (either year or second) between the weight values and the variable values for comparisons?
**Response:** We have chosen to report the weights in this manner in order to be consistent with the previous publications Cheng et al., (2024, 2025) and the PINNICLE documentation. The typical values are reported in m y$^{-1}$ as this is easily understood by members of the cryosphere community and the ice is moving too slowly for the velocities to be reported in m s$^{-1}$. The weight values are reported in the SI units as these are the real values used in the computations.

Line 167: Regarding "the PINN output variables will have different values for different regions of the GrIS", are you using the same weights across the three different regions where the velocities can be very different?
**Response:** The weight values are the same for all thee regions. This framework generally works since the mean velocities over these regions are generally comparable since they include both fast-flowing and slower-moving ice. Moreover, we also included a logarithmic terms in our loss function to account for the differences in the velocities across the regions. Deception and Narssap generally have slower velocities than Upernavik, and these logarithm loss terms help better captures the slower velocities. We will make this more clear in the methods section.

Line 172: I like the fact that different random seeds allow you to sample different solutions that can solve the ill-posed inverse problem. Given the ill-poseness, and the importance of training an ensemble of PINNs,

could you elaborate on why 5 time is sufficient, and if you expect different medians if you can train more PINNs?

**Response:** While increasing the number of PINNs in our ensemble is always desirable, we decided that 5 PINNs would be suitable for each region given the computational cost of training each PINN. We find that since we have chosen a high number of iterations for training each PINN in the ensemble, the PINN predictions generally converge and hence an ensemble of 5 PINNs is suitable.

Eqn 12: Why do you not need data loss for surface elevation s in the loss?

**Response:** The mass balance equation does not include a surface elevation term, hence we do not need the PINN to predict surface elevation values. As a result, we do not need to include this term in the loss function.

Line 205: Regarding "PINN (SB) is exposed to the ice velocity data along the boundaries of the region of interest, thereby satisfying the stress balance boundary conditions" How many velocity data points do you have along the boundary? I thought the velocity data is 400 points sampled within the domain, meaning that along the boundary of the ROI the velocity data points would be sparse, not truly "satisfying the stress balance boundary conditions".

**Response:** Thanks for pointing this out. We will rephrase this sentence to make it more clear within the manuscript. We do randomly select 4000 data points for $v_x$ and 4000 data points for $v_y$, and these include both points within the domain of interest as well as on the boundaries of the domain. It's true that most of these points will likely fall within the domain as opposed to along the boundaries. Using PINNICLE, we did not explicitly separate boundary points from the interior of the domain.

Additionally, as you're also solving for H and s, can you comment on the boundary conditions involving s and H? As they are within spatial derivatives in the momentum equation they also theoretically require BCs just like velocities.

**Response:** Similarly, we randomly select 4000 surface elevation and ice thickness data points from the training data sets. The domain boundary of surface is included as the velocity, however, there is no explicit domain boundary conditions for the ice thickness, because we are only using data along ice-penetrating radar flight tracks which do not usually follow the domain boundary. We assume that the ice thickness predictions are constrained as long as we have some ice thickness data along each flow line (Morlighem et al., 2011).

Somewhere in the paper write down $\sigma_{SSA}$ in stress balance in terms of velocities; this will make the discussions of the velocity boundary conditions more clear.

**Response:** Thank you for the suggestion, we will make this change in the manuscript.

Figure 4 d-f: Does positive denote higher Bedmachine or PINN topography?

**Response:** We will definitely make this more explicit in the Fig. 4 and Fig. 5 captions. We will state, explicitly, that the difference is $\hat{b}_{\text{BedMachine}} - \hat{b}$ for panels d-f.

Line 259: Regarding "the PINN (MB+SB) RMSE of 137 m is 260 lower than that of BedMachine (see Table 2)." I think you really need error bars for the PINN RMSE to compare which one is lower.

**Response:** Thank you for this feedback. We will include additional information on the uncertainties and error estimates in an updated version of this manuscript.

Line 278: Regarding "Lastly, the PINN predicts a 'disconnected' trough beneath the southern fork of Upernavik Isstrøm North, while BedMachine suggests that this trough is continuous." Do you also see disconnected troughs at each PINN result, prior to the averaging across PINN results? When taking the mean between different PINN results do you remove some features that were apparent in each individual PINN results?

**Response:** This is a good point that will likely become clear once we plot figures of the range/standard error of ensemble predictions for the bed topography in the Upernavik region. While it is possible that we are excluding some information by taking the median across PINN predictions, in almost all the experiments we have performed, the PINN has not been able to recover the full trough beneath the southern fork of Upernavik Isstrøm North. That being said, the 'disconnected' part of the trough is shaded dark green which is very close to a bed elevation of 0 m (Figure 5a), indicating that while the PINN does not predict the full

depth of the trough (as shown in BedMachine), it is quite close. We will add this point to our discussion!

Table 4: Why is thickness RMSE not included?
**Response:** The thickness (or bed topography) RMSEs are included in Tables 2 and 3. Table 4 is only meant for the other state variables in the PDEs that we are not trying to infer, i.e., velocity, apparent mass balance, and surface elevation.